# SCINet: Time Series Modeling and Forecasting with Sample Convolution and Interaction

**Minhao Liu**\*, **Ailing Zeng, Muxi Chen, Zhijian Xu, Qiuxia Lai, Lingna Ma, Qiang Xu**\*
CUhk REliable Computing (CURE) Lab.
Dept. of Computer Science & Egnineering, The Chinese University of Hong Kong
\*{mhliu,qxu}@cse.cuhk.edu.hk

## Abstract

One unique property of time series is that the temporal relations are largely preserved after downsampling into two sub-sequences. By taking advantage of this property, we propose a novel neural network architecture that conducts sample convolution and interaction for temporal modeling and forecasting, named **SCINet**. Specifically, SCINet is a recursive *downsample-convolve-interact* architecture. In each layer, we use multiple convolutional filters to extract *distinct yet valuable* temporal features from the downsampled sub-sequences or features. By combining these rich features aggregated from multiple resolutions, SCINet effectively models time series with complex temporal dynamics. Experimental results show that SCINet achieves significant forecasting accuracy improvements over both existing convolutional models and Transformer-based solutions across various real-world time series forecasting datasets. Our codes and data are available at `https://github.com/cure-lab/SCINet`.

## 1  Introduction

Time series forecasting (TSF) enables decision-making with the estimated future evolution of metrics or events, thereby playing a crucial role in various scientific and engineering fields such as healthcare [1], energy management [42], traffic flow [42], and financial investment [10], to name a few.

There are mainly three kinds of deep neural networks used for sequence modeling, and they are all applied for time series forecasting [24]: (i). recurrent neural networks (RNNs) [13]; (ii). Transformer-based models [37]; and (iii). temporal convolutional networks (TCN) [4].

Despite the promising results of TSF methods based on these generic models, they do not consider the specialty of time series data during modeling. For example, one unique property of time series is that the temporal relations (e.g., the trend and the seasonal components of the data) are largely preserved after downsampling into two sub-sequences. Consequently, by recursively downsampling the time series into sub-sequences, we could obtain a rich set of convolutional filters to extract dynamic temporal features at multiple resolutions.

Motivated by the above, in this paper, we propose a novel neural network architecture for time series modeling and forecasting, named *sample convolution and interaction network* (*SCINet*). The main contributions of this paper are as follows:

- We propose SCINet, a hierarchical *downsample-convolve-interact* TSF framework that effectively models time series with complex temporal dynamics. By iteratively extracting and exchanging information at multiple temporal resolutions, an effective representation with enhanced predictability can be learned, as verified by its comparatively lower permutation entropy (PE) [16].

36th Conference on Neural Information Processing Systems (NeurIPS 2022).

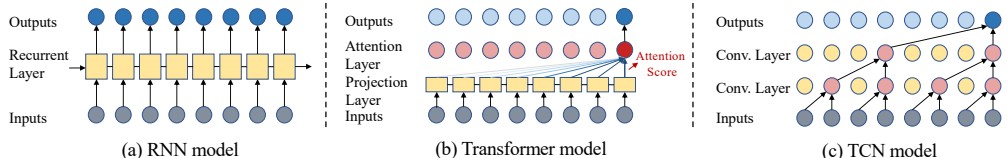

Figure 1: Existing sequence modeling architectures for time series forecasting.

- We design the basic building block, *SCI-Block*, for constructing SCINet, which downsamples the input data/feature into two sub-sequences, and then extracts features of each sub-sequence using distinct convolutional filters. To compensate for the information loss during the downsampling procedure, we incorporate interactive learning between the two convolutional features within each SCI-Block.

Extensive experiments on various real-world TSF datasets show that our model consistently outperforms existing TSF approaches by a considerable margin. Moreover, while SCINet does not explicitly model spatial relations, it achieves competitive forecasting accuracy on spatial-temporal TSF tasks.

## 2 Related Work and Motivation

The time series forecasting problem is defined as: Given a long time series $\mathbf{X}^*$ and a look-back window of fixed length $T$, at timestamp $t$, time series forecasting is to predict $\hat{\mathbf{X}}_{t+1:t+\tau} = \{\mathbf{x}_{t+1}, ..., \mathbf{x}_{t+\tau}\}$ based on the past $T$ steps $\mathbf{X}_{t-T+1:t} = \{\mathbf{x}_{t-T+1}, ..., \mathbf{x}_t\}$. Here, $\tau$ is the length of the forecast horizon, $\mathbf{x}_t \in \mathbb{R}^d$ is the value at time step $t$, and $d$ is the number of variates. For simplicity, in the following we will omit the subscripts, and use $\mathbf{X}$ and $\hat{\mathbf{X}}$ to represent the historical data and the forecasted data, respectively.

### 2.1 Related Work

Traditional time series forecasting methods such as the autoregressive integrated moving average (ARIMA) model [8] and Holt-Winters seasonal method [14] have theoretical guarantees. However, they are mainly applicable for univariate forecasting problems, restricting their applications to complex time series data. With the increasing data availability and computing power in recent years, it is shown that deep learning-based TSF techniques have the potential to achieve better forecasting accuracy than conventional approaches [24, 29].

Earlier RNN-based TSF methods [31, 32] summarize the past information compactly in the internal memory states that are recursively updated with new inputs at each time step, as shown in Fig. 1(a). The gradient vanishing/exploding problems and the inefficient training procedure greatly restrict the application of RNN-based models.

In recent years, Transformer-based models [37] have taken the place of RNN models in almost all sequence modeling tasks, thanks to the effectiveness and efficiency of the self-attention mechanisms. Various Transformer-based TSF methods (see Fig. 1(b)) are proposed in the literature [21, 23, 38, 25]. These works typically focus on the challenging long-term time series forecasting problem, taking advantage of their remarkable long sequence modeling capabilities.

Another popular type of TSF model is the so-called temporal convolutional network [7, 4, 33, 39, 27], wherein convolutional filters are used to capture local temporal features (see Fig. 1(c)). The proposed SCINet is also constructed based on temporal convolution. However, our method has several key differences compared with the TCN model based on dilated causal convolution, as discussed in the following.

### 2.2 Rethinking Dilated Causal Convolution for Time Series Modeling and Forecasting

The local correlation of time series data is reflected in the continuous changes within a time slot, and convolutional filters can effectively capture such local features. Consequently, convolutional neural networks are explored in the literature for time series modeling and forecasting. In particular, dilated causal convolution (DCS) is the current *de facto* method used in this respect.

DCS was first proposed for generating raw audio waveforms in WaveNet [28]. Later, [4] simplifies the WaveNet architecture to the so-called temporal convolutional networks (see Fig. 1 (c)). TCN consists of a stack of causal convolutional layers with exponentially enlarged dilation factors, which can achieve a large receptive field with just a few convolutional layers. Over the years, TCN has been widely used in all kinds of time series forecasting problems and achieve promising results [39, 33]. Moreover, convolutional filters can work seamlessly with graph neural networks (GNNs) to solve various spatial-temporal TSF problems.

With *causal convolutions* in the TCN architecture, an output $i$ is convolved only with the $i^{th}$ and earlier elements in the previous layer. While causality should be kept in forecasting tasks, the potential "future information leakage" problem exists only when the output and the input have temporal overlaps. In other words, causal convolutions should be applied only in autoregressive forecasting, wherein the previous output serves as the input for future prediction. When the predictions are completely based on the known inputs in the look-back window, there is no need to use causal convolutions. We can safely apply *normal convolutions* on the look-back window for forecasting.

More importantly, the dilated architecture in TCN has two inherent limitations:

- A single convolutional filter is shared within each layer. Such a unified convolutional kernel tends to extract the average temporal features from the data/features in the previous layer. However, complex time series may contain substantial temporal dynamics. Hence, it is essential to extract distinct yet valuable features with a rich set of convolutional filters.
- While the final layer of the TCN model has the global view of the entire look-back window, the effective receptive fields of the intermediate layers (especially those close to the inputs) are limited, causing temporal relation loss during feature extraction.

The above limitations of the TCN architecture motivate the proposed SCINet design, as detailed in the following section.

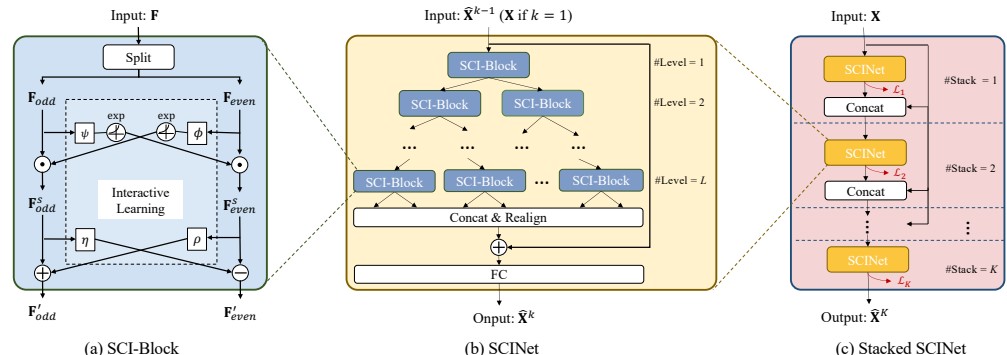

(a) SCI-Block          (b) SCINet          (c) Stacked SCINet

Figure 2: The overall architecture of Sample Convolution and Interaction Network (SCINet).

## 3 SCINet: Sample Convolution and Interaction Network

SCINet adopts an encoder-decoder architecture. The encoder is a hierarchical convolutional network that captures dynamic temporal dependencies at multiple resolutions with a rich set of convolutional filters. As shown in Fig. 2(a), the basic building block, *SCI-Block* (Section 3.1), downsamples the input data or feature into two sub-sequences and then processes each sub-sequence with a set of convolutional filters to extract distinct yet valuable temporal features from each part. To compensate for the information loss during downsampling, we incorporate *interactive learning* between the two sub-sequences. Our *SCINet* (Section 3.2) is constructed by arranging multiple SCI-Blocks into a binary tree structure (Fig. 2(b)). A distinctive advantage of such design is that each SCI-Block has both local and global views of the entire time series, thereby facilitating the extraction of useful temporal features. After all the downsample-convolve-interact operations, we realign the extracted features into a new sequence representation and add it to the original time series for forecasting with a fully-connected network as the decoder. To facilitate extracting complicated temporal patterns, we could further stack multiple SCINets and apply intermediate supervision to get a *Stacked SCINet* (Section 3.3), as shown in Fig. 2(c).

### 3.1 SCI-Block

The SCI-Block (Fig. 2(a)) is the basic module of the SCINet, which decomposes the input feature $\mathbf{F}$ into two sub-features $\mathbf{F}'_{odd}$ and $\mathbf{F}'_{even}$ through the operations of *Splitting* and *Interactive-learning*.

The *Splitting* procedure downsamples the original sequence $\mathbf{F}$ into two sub-sequences $\mathbf{F}_{even}$ and $\mathbf{F}_{odd}$ by separating the even and the odd elements, which are of coarser temporal resolution but preserve most information of the original sequence.

Next, we use different convolutional kernels to extract features from $\mathbf{F}_{even}$ and $\mathbf{F}_{odd}$. As the kernels are separate, the extracted features from them would contain distinct yet valuable temporal relations with enhanced representation capabilities. To compensate for potential information loss with downsampling, we propose a novel *interactive-learning* strategy to allow information interchange between the two sub-sequences by learning affine transformation parameters from each other. As shown in Fig. 2 (a), the interactive learning procedure consists of two steps.

First, $\mathbf{F}_{even}$ and $\mathbf{F}_{odd}$ are projected to hidden states with two different 1D convolutional modules $\phi$ and $\psi$, respectively, and transformed to the formats of $\exp$ and interact to the $\mathbf{F}_{even}$ and $\mathbf{F}_{odd}$ with the element-wise product (see Eq. (1)). This can be viewed as performing scaling transformation on $\mathbf{F}_{even}$ and $\mathbf{F}_{odd}$, where the scaling factors are learned from each other using neural network modules. Here, $\odot$ is the Hadamard product or element-wise production.

$$\mathbf{F}^s_{odd} = \mathbf{F}_{odd} \odot \exp(\phi(\mathbf{F}_{even})), \quad \mathbf{F}^s_{even} = \mathbf{F}_{even} \odot \exp(\psi(\mathbf{F}_{odd})). \tag{1}$$

$$\mathbf{F}'_{odd} = \mathbf{F}^s_{odd} \pm \rho(\mathbf{F}^s_{even}), \quad \mathbf{F}'_{even} = \mathbf{F}^s_{even} \pm \eta(\mathbf{F}^s_{odd}). \tag{2}$$

Second, as shown in Eq. (11), the two scaled features $\mathbf{F}^s_{even}$ and $\mathbf{F}^s_{odd}$ are further projected to another two hidden states with the other two 1D convolutional modules $\rho$ and $\eta$, and then added to or subtracted from[1] $\mathbf{F}^s_{even}$ and $\mathbf{F}^s_{odd}$. The final outputs of the interactive learning module are two updated sub-features $\mathbf{F}'_{even}$ and $\mathbf{F}'_{odd}$. The default architectures of $\phi$, $\psi$, $\rho$ and $\eta$ are shown in the Appendix C.

Compared to the dilated convolutions used in the TCN architecture, the proposed downsample-convolve-interact architecture achieves an even larger receptive field at each convolutional layer. More importantly, unlike TCN that employs a single shared convolutional filter at each layer, significantly restricting its feature extraction capabilities, SCI-Block aggregates essential information extracted from the two downsampled sub-sequences that have both local and global views of the entire time series.

### 3.2 SCINet

With the SCI-Blocks presented above, we construct the SCINet by arranging multiple SCI-Blocks hierarchically and get a tree-structured framework, as shown in Fig. 2 (b).

There are $2^l$ SCI-Blocks at the $l$-th level, where $l = 1, \ldots, L$ is the index of the level, and $L$ is the total number of levels. Within the $k$-th SCINet of the stacked SCINet (Section 3.3), the input time series $\mathbf{X}$ (for $k=1$) or feature vector $\hat{\mathbf{X}}^{k-1} = \{\hat{\mathbf{x}}^{k-1}_1, \ldots, \hat{\mathbf{x}}^{k-1}_\tau\}$ (for $k>1$) is gradually down-sampled and processed by SCI-Blocks through different levels, which allows for effective feature learning of different temporal resolutions. In particular, the information from previous levels will be gradually accumulated, i.e., the features of the deeper levels would contain extra finer-scale temporal information transmitted from the shallower levels. In this way, we can capture both short-term and long-term temporal dependencies in the time series.

After going through $L$ levels of SCI-Blocks, we rearrange the elements in all the sub-features by reversing the odd-even splitting operation and concatenate them into a new sequence representation. It is then added to the original time series through a residual connection [12] to generate a new sequence with enhanced predictability. Finally, a simple fully-connected network is used to decode the enhanced sequence representation into $\hat{\mathbf{X}}^k = \{\hat{\mathbf{x}}^k_1, \ldots, \hat{\mathbf{x}}^k_\tau\}$. Note that, to mitigate distribution shift in some TSF tasks, before supplying the data in the look-back window to our model, all the data elements are subtracted with the value of the last element, which is added to all the data elements in the forecasting horizon afterwards.

---

[1]The selection of the operators in Eq.(2) affects the parameter initialization of our model and we show its impact in the Appendix B.3.

### 3.3 Stacked SCINet

When there are sufficient training samples, we could stack $K$ layers of SCINets to achieve even better forecasting accuracy (see Fig. 2 (c)), at the cost of a more complex model structure.

Specifically, we apply *intermediate supervision* [5] on the output of each SCINet using the ground-truth values, to ease the learning of the intermediate temporal features. The output of the $k$-th intermediate SCINet, $\hat{\mathbf{X}}^k$ with length $\tau$, is concatenated with part of the input $\mathbf{X}_{t-(T-\tau)+1:t}$ to recover the length to the original input and feeded as input into the $(k+1)$-th SCINet, where $k = 1, \ldots, K-1$, and $K$ is the total number of the SCINets in the stacked structure. The output of the $K$-th SCINet, $\hat{\mathbf{X}}^K$, is the final forecasting results.

### 3.4 Loss Function

To train a stacked SCINet with $K$ ($K \geq 1$) SCINets, the loss of the $k$-th prediction results is calculated as the L1 loss between the output of the $k$-th SCINet and the ground-truth horizontal window to be predicted:

$$\mathcal{L}_k = \frac{1}{\tau} \sum_{i=0}^{\tau} \left\| \hat{\mathbf{x}}_i^k - \mathbf{x}_i \right\| \tag{3}$$

The total loss of the stacked SCINet can be written as:

$$\mathcal{L} = \sum_{k=1}^{K} \mathcal{L}_k. \tag{4}$$

### 3.5 Complexity Analysis

Thanks to the downsampling procedure, the neurons at each convolutional layer of SCINet have a larger receptive field than those of TCN. More importantly, the set of rich convolutional filters in SCINet enable flexible extraction of temporal features from multiple resolutions. Consequently, SCINet usually does not require downsampling the original sequence to the coarsest level for effective forecasting. Given the look-back window size $T$, TCN generally requires $\lceil \log_2 T \rceil$ layers when the dilation factor is 2, while the number of layers $L$ in SCINet could be much smaller than $\log_2 T$. Our empirical study shows that the best forecasting accuracy is achieved with $L \leq 5$ in most cases even with large $T$ (e.g., 168). As for the number of stacks $K$, our empirical study also shows that $K \leq 3$ would be sufficient.

Consequently, the computational cost of SCINet is usually on par with that of the TCN architecture. The worst-case time complexity is $\mathcal{O}(T \log T)$, much less than that of vanilla Transformer-based solutions: $\mathcal{O}(T^2)$.

## 4 Experiments

In this section, we show the quantitative and qualitative comparisons with the state-of-the-art models for time series forecasting. We also present a comprehensive ablation study to evaluate the effectiveness of different components in SCINet. More details on *datasets*, *evaluation metrics*, *data pre-processing*, *experimental settings*, *network structures* and their *hyper-parameters* are shown in the Appendix.

### 4.1 Datasets

We conduct experiments on 11 popular time series datasets: (1) *Electricity Transformer Temperature* [42] (ETTh) (2) *Traffic* (3) *Solar-Energy* (4) *Electricity* (5) *Exchange-Rate* (6) *PeMS* (*PEMS03, PEMS04, PEMS07 and PEMS08*). A brief description of these datasets is listed in Table 1. All the experiments on these datasets in this section are conducted under multi-variate TSF setting.

To make a fair comparison, we follow existing experimental settings, and use the same evaluation metrics as the original publications [17, 26, 40, 19] in each dataset.

Table 1: The overall information of the 11 datasets.

| Datasets | ETTh (1,2) | ETTm1 | Traffic | Solar-Energy | Electricity | Exchange-Rate | PEMS03 | PEMS04 | PEMS07 | PEMS08 |
|---|---|---|---|---|---|---|---|---|---|---|
| Variants | 7 | 7 | 862 | 137 | 321 | 8 | 358 | 307 | 883 | 170 |
| Timesteps | 17,420 | 69,680 | 17,544 | 52,560 | 26,304 | 7,588 | 26,209 | 16,992 | 28,224 | 17,856 |
| Granularity | 1hour | 15min | 1hour | 10min | 1hour | 1day | 5min | 5min | 5min | 5min |
| Start time | 7/1/2016 | 7/1/2016 | 1/1/2015 | 1/1/2006 | 1/1/2012 | 1/1/1990 | 5/1/2012 | 7/1/2017 | 5/1/2017 | 3/1/2012 |
| Task type | Multi-step | Multi-step | Single-step | Single-step | Single-step | Single-step | Multi-step | Multi-step | Multi-step | Multi-step |
| Data partition | Follow [42] | | Training/Validation/Testing: 6/2/2 | | | | Training/Validation/Testing: 6/2/2 | | | |

## 4.2 Results and Analyses

Table 2, 3, 4, 5, 6 provide the main experimental results of SCINet. We observe that SCINet shows superior performance than other TSF models on various tasks, including short-term, long-term and spatial-temporal time series forecasting.

**Short-term Time Series Forecasting:** we evaluate the performance of the SCINet in short-term TSF tasks with other baseline methods on *Traffic*, *Solar-Energy*, *Electricity* and *Exchange-Rate* datasets. The experimental setting is the same as [19], which uses the input length of 168 to forecast different future horizons$\{3, 6, 12, 24\}$.

As can be seen in Table 2, the proposed SCINet outperforms existing RNN/TCN-based (LSTNet [19], TPA-LSTM [34], TCN [4], TCN†) and Transformer-based [38, 42, 37] TSF solutions in most cases, especially for the Solar-Energy and Exchange-Rate datasets. Note that, TCN† denotes a variant of TCN wherein causal convolutions are replaced by normal convolutions, and improves the original TCN across all the datasets, which supports our claim in Sec. 2.2. Moreover, we can also observe that the Transformer-based methods have poor performance in this task. For short-term forecasting, the recent data points are typically more important for accurate forecasting. However, the permutation-invariant self-attention mechanisms used in Transformer-based methods do not pay much attention to such critical information. In contrast, the general sequential models (RNN/TCN) can formulate it easily, showing quite competitive results in short-term forecasting.

Table 2: Short-term forecasting performance comparison on the four datasets. The best results are shown in **bold** and second best results are highlighted with underlined blue font. IMP shows the improvement of SCINet over the best model.

| Model | | SCINet | | Autoformer [40] | | Informer [42] | | Transformer [37] | | *TCN [4] | | *TCN† | | LSTNet [19] | | TPA-LSTM [34] | | IMP |
|---|---|---|---|---|---|---|---|---|---|---|---|---|---|---|---|---|---|---|
| Metric | τ | RSE | CORR | RSE | CORR | RSE | CORR | RSE | CORR | RSE | CORR | RSE | CORR | RSE | CORR | RSE | CORR | RSE |
| Solar-Energy | 3 | **0.1775** | **0.9853** | N/A | N/A | N/A | N/A | N/A | N/A | 0.1940 | 0.9835 | 0.1900 | 0.9848 | 0.1843 | 0.9843 | 0.1803 | 0.9850 | 1.55% |
| | 6 | **0.2301** | **0.9739** | N/A | N/A | N/A | N/A | N/A | N/A | 0.2581 | 0.9602 | 0.2382 | 0.9612 | 0.2559 | 0.9690 | 0.2347 | 0.9742 | 1.96% |
| | 12 | **0.2997** | **0.9550** | N/A | N/A | N/A | N/A | N/A | N/A | 0.3512 | 0.9321 | 0.3353 | 0.9432 | 0.3254 | 0.9467 | 0.3234 | 0.9487 | 7.33% |
| | 24 | **0.4081** | **0.9112** | N/A | N/A | N/A | N/A | N/A | N/A | 0.4732 | 0.8812 | 0.4676 | 0.8851 | 0.4643 | 0.8870 | 0.4389 | 0.9081 | 7.02% |
| Traffic | 3 | **0.4216** | **0.8920** | 0.5368 | 0.8268 | 0.5175 | 0.8515 | 0.5122 | 0.8555 | 0.5459 | 0.8486 | 0.5361 | 0.8540 | 0.4777 | 0.8721 | 0.4487 | 0.8812 | 6.04% |
| | 6 | **0.4414** | **0.8809** | 0.5462 | 0.8191 | 0.5258 | 0.8465 | 0.5455 | 0.8388 | 0.6061 | 0.8205 | 0.5992 | 0.8197 | 0.4893 | 0.8690 | 0.4658 | 0.8717 | 5.24% |
| | 12 | **0.4495** | **0.8772** | 0.5623 | 0.8082 | 0.5533 | 0.8279 | 0.5485 | 0.8317 | 0.6367 | 0.8048 | 0.6061 | 0.8205 | 0.4950 | 0.8614 | 0.4641 | 0.8717 | 3.15% |
| | 24 | **0.4453** | **0.8825** | 0.6020 | 0.7757 | 0.5883 | 0.8033 | 0.5934 | 0.8048 | 0.6586 | 0.7921 | 0.6456 | 0.7982 | 0.4973 | 0.8588 | 0.4765 | 0.8629 | 6.55% |
| Electricity | 3 | **0.0740** | **0.9494** | 0.1458 | 0.9032 | 0.1524 | 0.8858 | 0.1182 | 0.9055 | 0.0892 | 0.9232 | 0.0852 | 0.9293 | 0.0864 | 0.9283 | 0.0823 | 0.9439 | 10.09% |
| | 6 | **0.0845** | **0.9387** | 0.1555 | 0.8957 | 0.1932 | 0.8660 | 0.1328 | 0.8962 | 0.0974 | 0.9121 | 0.0924 | 0.9235 | 0.0931 | 0.9135 | 0.0916 | 0.9337 | 7.75% |
| | 12 | **0.0929** | **0.9305** | 0.1541 | 0.8907 | 0.1748 | 0.8585 | 0.1375 | 0.8849 | 0.1053 | 0.9017 | 0.0993 | 0.9173 | 0.1007 | 0.9077 | 0.0964 | 0.9250 | 3.63% |
| | 24 | **0.0967** | **0.9270** | 0.1754 | 0.8732 | 0.2110 | 0.8347 | 0.1461 | 0.8774 | 0.1091 | 0.9101 | 0.0989 | 0.9101 | 0.1007 | 0.9119 | 0.1006 | 0.9133 | 3.88% |
| Exchange Rate | 3 | **0.0171** | **0.9787** | 0.0400 | 0.9458 | 0.1392 | 0.9473 | 0.0689 | 0.9759 | 0.0217 | 0.9693 | 0.0202 | 0.9712 | 0.0226 | 0.9735 | 0.0174 | 0.979 | 1.72% |
| | 6 | **0.0240** | 0.9704 | 0.0481 | 0.9197 | 0.1548 | 0.9207 | 0.0806 | 0.9671 | 0.0263 | 0.9633 | 0.0257 | 0.9628 | 0.0280 | 0.9658 | 0.0241 | **0.9709** | 0.41% |
| | 12 | **0.0331** | 0.9553 | 0.0638 | 0.9054 | 0.1793 | 0.8817 | 0.0893 | 0.9476 | 0.0393 | 0.9531 | 0.0352 | 0.9501 | 0.0356 | 0.9511 | 0.0341 | **0.9564** | 2.93% |
| | 24 | **0.0436** | **0.9396** | 0.0651 | 0.8952 | 0.1998 | 0.7715 | 0.1127 | 0.9213 | 0.0492 | 0.9223 | 0.0487 | 0.9314 | 0.0449 | 0.9354 | 0.0444 | 0.9381 | 1.80% |

- Autoformer, Informer and Transformer achieved by Autoformer [40] requires pre-prossessed datasets for training.
- N/A denotes no pre-prossessed dataset for training.
- ∗ denotes re-implementation.  † denotes the variant with normal convolutions.

**Long-term Time Series Forecasting:** many real-world applications also require to predict long-term events. Therefore, we conduct the experiments on *Exchange Rate, Electricity ,Traffic* and *ETT* datasets to evaluate the performance of SCINet on long-term TSF tasks. In this experiment, we only compare SCINet with Transformer-based methods [38, 18, 21, 42, 37, 25], since they are more popular in recent long-term TSF research.

As can be seen from Table 3, the SCINet achieves state-of-the-art performances in most benchmarks and prediction length settings. Overall, SCINet yields 39.89% average improvements on MSE among the above settings. In particular, for Exchange-Rate, compared to previous state-of-the-art results, SCINet gives average 65% improvements on MSE. We attribute it to that the proposed SCINet can better capture both short (*local temporal dynamics*)- and long (*trend, seasonality*)-term temporal dependencies to make an accurate prediction in long-term TSF.

Table 3: Long-term forecasting performance comparison with Transformer-based models.

| Model | | SCINet | | Autoformer [38] | | *Pyraformer [25] | | Informer [42] | | Transformer [37] | | LogTrans [21] | | Reformer [18] | | IMP |
|---|---|---|---|---|---|---|---|---|---|---|---|---|---|---|---|---|
| Metric | | MSE | MAE | MSE | MAE | MSE | MAE | MSE | MAE | MSE | MAE | MSE | MAE | MSE | MAE | MSE |
| Exchange Rate | 96 | **0.061** | **0.188** | 0.197 | 0.323 | 1.748 | 1.105 | 0.847 | 0.752 | 0.559 | 0.587 | 0.968 | 0.812 | 1.065 | 0.829 | 68.98% |
| | 192 | **0.106** | **0.244** | 0.300 | 0.369 | 1.874 | 1.151 | 1.204 | 0.895 | 1.168 | 0.835 | 1.040 | 0.851 | 1.188 | 0.906 | 64.70% |
| | 336 | **0.181** | **0.323** | 0.509 | 0.524 | 1.943 | 1.172 | 1.672 | 1.036 | 1.423 | 0.949 | 1.659 | 1.081 | 1.357 | 0.976 | 64.36% |
| | 720 | **0.525** | **0.571** | 1.447 | 0.941 | 2.085 | 1.206 | 2.478 | 2.478 | 2.160 | 1.150 | 1.941 | 1.127 | 1.510 | 1.016 | 63.72% |
| Electricity | 96 | **0.168** | **0.253** | 0.201 | 0.317 | 0.386 | 0.449 | 0.274 | 0.368 | 0.263 | 0.359 | 0.258 | 0.357 | 0.312 | 0.402 | 16.42% |
| | 192 | **0.175** | **0.262** | 0.222 | 0.334 | 0.378 | 0.443 | 0.296 | 0.296 | 0.273 | 0.374 | 0.266 | 0.368 | 0.348 | 0.433 | 21.17% |
| | 336 | **0.189** | **0.278** | 0.231 | 0.338 | 0.376 | 0.443 | 0.300 | 0.394 | 0.277 | 0.373 | 0.280 | 0.380 | 0.350 | 0.433 | 18.19% |
| | 720 | **0.231** | **0.316** | 0.254 | 0.361 | 0.376 | 0.445 | 0.373 | 0.439 | 0.290 | 0.378 | 0.283 | 0.376 | 0.340 | 0.420 | 9.06% |
| Traffic | 96 | **0.613** | 0.395 | **0.613** | 0.388 | 0.867 | 0.468 | 0.719 | 0.391 | 0.638 | 0.354 | 0.684 | 0.384 | 0.732 | 0.423 | 0.00% |
| | 192 | **0.535** | **0.355** | 0.616 | 0.382 | 0.869 | 0.467 | 0.696 | 0.379 | 0.647 | 0.354 | 0.685 | 0.390 | 0.733 | 0.420 | 13.15% |
| | 336 | **0.540** | **0.359** | 0.622 | 0.337 | 0.881 | 0.469 | 0.777 | 0.420 | 0.669 | 0.364 | 0.733 | 0.408 | 0.742 | 0.420 | 13.18% |
| | 720 | **0.620** | **0.394** | 0.660 | 0.408 | 0.896 | 0.473 | 0.864 | 0.472 | 0.707 | 0.386 | 0.717 | 0.396 | 0.755 | 0.423 | 6.06% |

- ∗ denotes re-implementation.

Table 4: Multivariate time-series forecasting results on the *ETT* datasets.

| Methods | Metrics | ETTh1 Horizon | | | | | ETTh2 Horizon | | | | | ETTm1 Horizon | | | | |
|---|---|---|---|---|---|---|---|---|---|---|---|---|---|---|---|---|
| | | 24 | 48 | 168 | 336 | 720 | 24 | 48 | 168 | 336 | 720 | 24 | 48 | 96 | 288 | 672 |
| LogTrans [21] | MSE | 0.686 | 0.766 | 1.002 | 1.362 | 1.397 | 0.828 | 1.806 | 4.070 | 3.875 | 3.913 | 0.419 | 0.507 | 0.768 | 1.462 | 1.669 |
| | MAE | 0.604 | 0.757 | 0.846 | 0.952 | 1.291 | 0.750 | 1.034 | 1.681 | 1.763 | 1.552 | 0.412 | 0.583 | 0.792 | 1.320 | 1.461 |
| Reformer [18] | MSE | 0.991 | 1.313 | 1.824 | 2.117 | 2.415 | 1.531 | 1.871 | 4.660 | 4.028 | 5.381 | 0.724 | 1.098 | 1.433 | 1.820 | 2.187 |
| | MAE | 0.754 | 0.906 | 1.138 | 1.280 | 1.520 | 1.613 | 1.735 | 1.846 | 1.688 | 2.015 | 0.607 | 0.777 | 0.945 | 1.094 | 1.232 |
| LSTMa [2] | MSE | 0.650 | 0.702 | 1.212 | 1.424 | 1.960 | 1.143 | 1.671 | 4.117 | 3.434 | 3.963 | 0.621 | 1.392 | 1.339 | 1.740 | 2.736 |
| | MAE | 0.624 | 0.675 | 0.867 | 0.994 | 1.322 | 0.813 | 1.221 | 1.674 | 1.549 | 1.788 | 0.629 | 0.939 | 0.913 | 1.124 | 1.555 |
| LSTNet [19] | MSE | 1.293 | 1.456 | 1.997 | 2.655 | 2.143 | 2.742 | 3.567 | 3.242 | 2.544 | 4.625 | 1.968 | 1.999 | 2.762 | 1.257 | 1.917 |
| | MAE | 0.901 | 0.960 | 1.214 | 1.369 | 1.380 | 1.457 | 1.687 | 2.513 | 2.591 | 3.709 | L1700 | 1.215 | 1.542 | 2.076 | 2.941 |
| Informer [42] | MSE | 0.577 | 0.685 | 0.931 | 1.128 | 1.215 | 0.720 | 1.457 | 3.489 | 2.723 | 3.467 | 0.323 | 0.494 | 0.678 | 1.056 | 1.192 |
| | MAE | 0.549 | 0.625 | 0.752 | 0.873 | 0.896 | 0.665 | 1.001 | 1.515 | 1.340 | 1.473 | 0.369 | 0.503 | 0.614 | 0.786 | 0.926 |
| *TCN [4] | MSE | 0.511 | 0.515 | 0.694 | 0.814 | 0.944 | 0.444 | 0.617 | 2.405 | 2.486 | 2.608 | 0.229 | 0.239 | 0.260 | 0.768 | 2.732 |
| | MAE | 0.549 | 0.529 | 0.617 | 0.682 | 0.778 | 0.478 | 0.615 | 1.266 | 1.312 | 1.276 | 0.282 | 0.360 | 0.363 | 0.646 | 1.371 |
| *Pyraformer [25] | MSE | 0.479 | 0.518 | 0.758 | 0.891 | 0.963 | 0.477 | 0.934 | 3.913 | 0.907 | 0.963 | 0.332 | 0.492 | 0.543 | 0.656 | 0.901 |
| | MAE | 0.499 | 0.520 | 0.665 | 0.738 | 0.782 | 0.537 | 0.764 | 1.557 | 0.747 | 0.783 | 0.383 | 0.475 | 0.510 | 0.598 | 0.720 |
| Autoformer [38] | MSE | 0.406 | 0.478 | 0.493 | 0.515 | **0.499** | 0.260 | 0.311 | 0.466 | 0.472 | 0.480 | 0.408 | 0.499 | 0.540 | 0.636 | 0.699 |
| | MAE | 0.440 | 0.462 | 0.481 | 0.492 | **0.500** | 0.339 | 0.372 | 0.458 | 0.478 | 0.488 | 0.424 | 0.464 | 0.489 | 0.533 | 0.564 |
| SCINet | MSE | **0.300** | **0.361** | **0.408** | **0.504** | 0.544 | **0.180** | **0.230** | **0.342** | **0.365** | **0.475** | **0.106** | **0.136** | **0.165** | **0.253** | **0.346** |
| | MAE | **0.342** | **0.388** | **0.417** | **0.495** | 0.527 | **0.263** | **0.303** | **0.380** | **0.409** | **0.488** | **0.202** | **0.230** | **0.252** | **0.315** | **0.376** |
| IMP | MSE | 26.11% | 24.48% | 17.24% | 2.14% | -9.02% | 30.77% | 25.81% | 26.61% | 22.67% | 1.04% | 38.71% | 22.83% | 21.40% | 49.59% | 40.18% |

- ∗ denotes re-implementation.

We conduct both *Multivariate Time-series Forecasting* and *Univariate Time-series Forecasting* on *ETT* datasets [42]. For a fair comparison, we keep all input lengths $T$ the same as those of *Informer*. The results are shown in Table 4 and Table 5, respectively.

*Multivariate Time-series Forecasting on ETT*: as can be seen from Table 4, compared with RNN-based methods such as LSTMa [2] and LSTnet [19], Transformer-based methods [18, 21, 42] produce better forecasting results. One of the primary reasons is that, RNN-based solutions conduct iterative forecasting and it is inevitable to suffer from error accumulation effects. As another direct forecasting method, TCN further outperforms vanilla Transformer-based methods [18, 21, 42], because the stacked convolutional layers allow for more effective local-to-global temporal relation learning for multivariate time series. It is worth noting that SCINet outperforms all the above models by a large margin. Fig. 3 presents the qualitative results on some randomly selected sequences of the ETTh1 dataset, which clearly demonstrate the capability of SCINet in obtaining the trend and seasonality of time series for TSF.

*Univariate Time-series Forecasting on ETT*: in this experimental setting, we bring several strong baseline methods for univariate forecasting into comparison, including ARIMA, Prophet [36], DeepAR [32] and N-Beats [29]. In Table 5, we can observe that N-Beats is superior to other baseline methods in most cases. In fact, N-Beats also takes the unique properties of time series into consideration and directly learns a trend and a seasonality model using a deep stack of fully-connected layers with residuals, which is a departure from the predominant architectures, such as RNNs, CNNs and Transformers. Nevertheless, the performance of SCINet is still much better than N-Beats.

The newly-proposed Transformer-based forecasting model, Autoformer [38], achieves the second best performance in all experimental settings and also surpasses SCINet in ETTm1 when the forecasting horizon is large. This is because, on the one hand, Autoformer focuses on modeling seasonal patterns and conducts self-attention at the sub-series level (instead of the raw data), which is much better in extracting long-term temporal patterns than vanilla Transformer-based methods. On the other hand, when forecasting long horizons, it is often the trend/seasonal information instead of the temporal dynamics in the look-back window that play the primary role, wherein the advantages of SCINet are not fully exhibited.

Table 5: Univariate time-series forecasting results on the *ETT* datasets.

| Methods | Metrics | ETTh1 | | | | | ETTh2 | | | | | ETTm1 | | | | |
|---|---|---|---|---|---|---|---|---|---|---|---|---|---|---|---|---|
| | | Horizon | | | | | Horizon | | | | | Horizon | | | | |
| | | 24 | 48 | 168 | 336 | 720 | 24 | 48 | 168 | 336 | 720 | 24 | 48 | 96 | 288 | 672 |
| ARIMA | MSE | 0.108 | 0.175 | 0.396 | 0.468 | 0.659 | 3.554 | 3.190 | 2.800 | 2.753 | 2.878 | 0.090 | 0.179 | 0.272 | 0.462 | 0.639 |
| | MAE | 0.284 | 0.424 | 0.504 | 0.593 | 0.766 | 0.445 | 0.474 | 0.595 | 0.738 | 1.044 | 0.206 | 0.306 | 0.399 | 0.558 | 0.697 |
| Prophet [36] | MSE | 0.115 | 0.168 | 1.224 | 1.549 | 2.735 | 0.199 | 0.304 | 2.145 | 2.096 | 3.355 | 0.120 | 0.133 | 0.194 | 0.452 | 2.747 |
| | MAE | 0.275 | 0.330 | 0.763 | 1.820 | 3.253 | 0.381 | 0.462 | 1.068 | 2.543 | 4.664 | 0.290 | 0.305 | 0.396 | 0.574 | 1.174 |
| DeepAR [32] | MSE | 0.107 | 0.162 | 0.239 | 0.445 | 0.658 | 0.098 | 0.163 | 0.255 | 0.604 | 0.429 | 0.091 | 0.219 | 0.364 | 0.948 | 2.437 |
| | MAE | 0.280 | 0.327 | 0.422 | 0.552 | 0.707 | 0.263 | 0.341 | 0.414 | 0.607 | 0.580 | 0.243 | 0.362 | 0.496 | 0.795 | 1.352 |
| N-Beats [29] | MSE | 0.042 | 0.065 | 0.106 | 0.127 | 0.269 | 0.078 | 0.123 | 0.244 | 0.270 | 0.281 | 0.031 | 0.056 | 0.095 | 0.157 | 0.207 |
| | MAE | 0.156 | 0.200 | 0.255 | 0.284 | 0.422 | 0.210 | 0.271 | 0.393 | 0.418 | 0.432 | 0.117 | 0.168 | 0.234 | 0.311 | 0.370 |
| Informer [42] | MSE | 0.098 | 0.158 | 0.183 | 0.222 | 0.269 | 0.093 | 0.155 | 0.232 | 0.263 | 0.277 | 0.030 | 0.069 | 0.194 | 0.401 | 0.512 |
| | MAE | 0.247 | 0.319 | 0.346 | 0.387 | 0.435 | 0.240 | 0.314 | 0.389 | 0.417 | 0.431 | 0.137 | 0.203 | 0.372 | 0.554 | 0.644 |
| Autoformer [38] | MSE | 0.057 | 0.103 | 0.090 | 0.106 | 0.120 | 0.110 | 0.123 | 0.188 | 0.225 | **0.257** | 0.025 | **0.039** | **0.057** | **0.103** | **0.110** |
| | MAE | 0.188 | 0.257 | 0.235 | 0.254 | 0.277 | 0.259 | 0.271 | 0.340 | 0.376 | **0.402** | 0.122 | **0.156** | **0.184** | **0.253** | **0.261** |
| SCINet | MSE | **0.029** | **0.041** | **0.071** | **0.084** | **0.099** | **0.065** | **0.093** | **0.158** | **0.166** | 0.286 | **0.019** | 0.045 | 0.064 | 0.111 | 0.165 |
| | MAE | **0.127** | **0.154** | **0.210** | **0.234** | **0.250** | **0.183** | **0.227** | **0.311** | **0.329** | 0.429 | **0.084** | 0.138 | 0.183 | 0.252 | 0.316 |
| **IMP** | MSE | 49.12% | 60.19% | 21.11% | 20.75% | 17.50% | 40.90% | 24.39% | 15.96% | 26.22% | -11.28% | 24.00% | -15.38% | -12.28% | -7.76% | -50.00% |

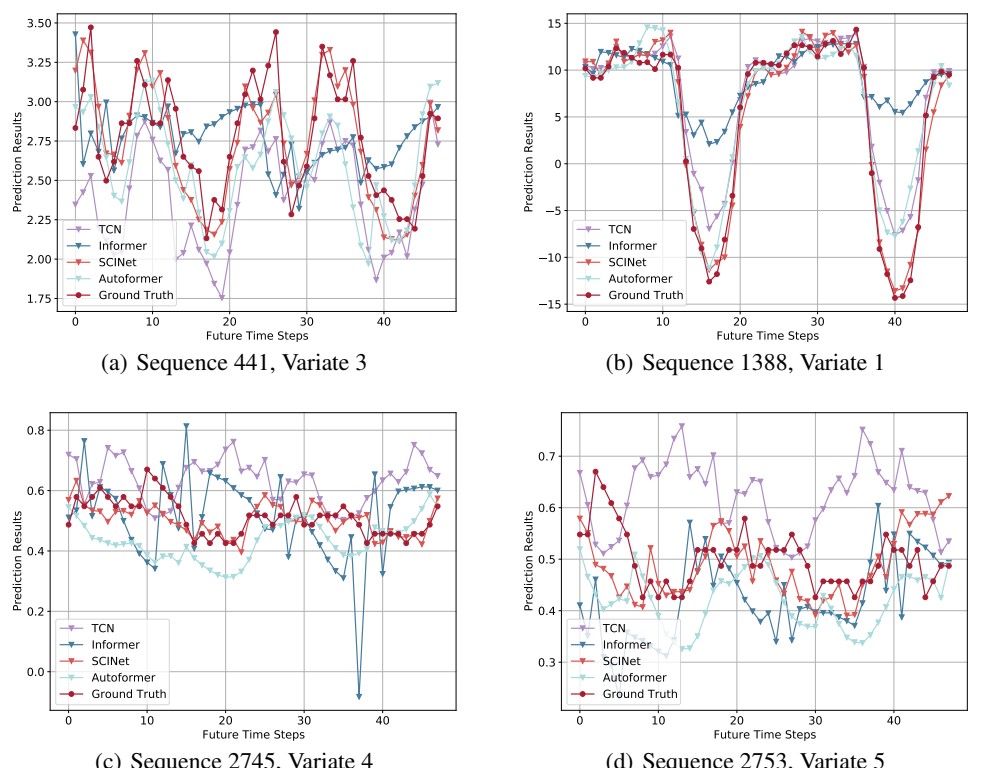

(a) Sequence 441, Variate 3

(b) Sequence 1388, Variate 1

(c) Sequence 2745, Variate 4

(d) Sequence 2753, Variate 5

Figure 3: The prediction results (Horizon = 48) of SCINet, Autoformer, Informer, and TCN on randomly-selected sequences from ETTh1 dataset.

**Spatial-temporal Time Series Forecasting:** besides the general TSF tasks, there is also a large category of data related to spatial-temporal forecasting. For example, traffic datasets *PeMS* [9] (PEMS03, PEMS04, PEMS07 and PEMS08) are complicated spatial-temporal time series for public traffic network and they have been investigated for decades. Most recent approaches: DCRNN [22], STGCN [41], ASTGCN [11], GraphWaveNet [39], STSGCN [35], AGCRN [3], LSGCN [15] and STFGNN [20] use graph neural networks to capture spatial relations while modeling temporal dependencies via conventional TCN or RNN/LSTM architectures. We follow the same experimental settings as the above works. As shown in Table 6, these GNN-based methods generally perform better than pure RNN or TCN-based methods. However, SCINet still achieves better performance without sophisticated spatial relation modeling, which further proves the superb temporal modeling capabilities of SCINet.

Table 6: Performance comparison of different approaches on the *PeMS* datasets.

| Datasets | Metrics | Methods | | | | | | | | | | | | IMP |
| | | *LSTM | *TCN | *TCN† | DCRNN | STGCN | ASTGCN(r) | GraphWaveNet | STSGCN | STFGNN | AGCRN | LSGCN | SCINet | MAE |
| --- | --- | --- | --- | --- | --- | --- | --- | --- | --- | --- | --- | --- | --- | --- |
| PEMS03 | MAE | 21.33 | 19.32 | 18.87 | 18.18 | 17.49 | 17.69 | 19.85 | 17.48 | 16.77 | *15.98 | - | **14.98** | 6.26% |
| | MAPE | 21.33 | 19.93 | 18.63 | 18.91 | 17.15 | 19.40 | 19.31 | 16.78 | 16.30 | *15.23 | - | **14.11** | 7.36% |
| | RMSE | 35.11 | 33.55 | 32.24 | 30.31 | 30.12 | 29.66 | 32.94 | 29.21 | 28.34 | *28.25 | - | **24.08** | 8.37% |
| PEMS04 | MAE | 25.14 | 23.22 | 22.81 | 24.70 | 22.70 | 22.93 | 25.45 | 21.19 | 19.83 | 19.83 | 21.53 | **18.95** | 4.44% |
| | MAPE | 20.33 | 15.59 | 14.31 | 17.12 | 14.59 | 16.56 | 17.29 | 13.90 | 13.02 | 12.97 | 13.18 | **11.86** | 8.56% |
| | RMSE | 39.59 | 37.26 | 36.87 | 38.12 | 35.55 | 35.22 | 39.70 | 33.65 | 31.88 | 32.30 | 33.86 | **30.89** | 4.40% |
| PEMS07 | MAE | 29.98 | 32.72 | 30.53 | 28.30 | 25.38 | 28.05 | 26.85 | 24.26 | 22.07 | *22.37 | - | **21.19** | 5.27% |
| | MAPE | 15.33 | 14.26 | 13.88 | 11.66 | 11.08 | 13.92 | 12.12 | 10.21 | 9.21 | *9.12 | - | **8.83** | 3.18% |
| | RMSE | 42.84 | 42.23 | 41.02 | 38.58 | 38.78 | 42.57 | 42.78 | 39.03 | 35.80 | *36.55 | - | **34.03** | 6.89% |
| PEMS08 | MAE | 22.20 | 22.72 | 21.42 | 17.86 | 18.02 | 18.61 | 19.13 | 17.13 | 16.64 | 15.95 | 17.73 | **15.72** | 1.44% |
| | MAPE | 15.32 | 14.03 | 13.09 | 11.45 | 11.40 | 13.08 | 12.68 | 10.96 | 10.60 | 10.09 | 11.20 | **9.80** | 2.87% |
| | RMSE | 32.06 | 35.79 | 34.03 | 27.83 | 27.83 | 28.16 | 31.05 | 26.80 | 26.22 | 25.22 | 26.76 | **24.76** | 1.82% |

- dash denotes that the methods do not implement on this dataset. ∗ denotes re-implementation or re-training. † denotes the variant with normal convolutions.

**Predictability estimation:** inspired by [16, 30], we use *permutation entropy* (*PE*) [6] to measure the predictability of the original input and the enhanced representation learnt by SCINet. Time series with lower PE values are regarded as less complex, thus theoretically easier to predict[2]. The PE values of the original time series and the corresponding enhanced representations are shown in Table 7.

Table 7: Permutation entropy comparison before and after SCINet.

| Permutation Entropy | | Datasets | | | | | | | | |
| | | ETTh1 | Traffic | Solar-Energy | Electricity | Exc-rate | PEMS03 | PEMS04 | PEMS07 | PEMS08 |
| --- | --- | --- | --- | --- | --- | --- | --- | --- | --- | --- |
| Parameters | m $(\tau = 1)$* | 6 | 6 | 7 | 6 | 6 | 6 | 6 | 6 | 6 |
| Value | Original Input | 0.8878 | 0.9371 | 0.4739 | 0.9489 | 0.8260 | 0.9649 | 0.9203 | 0.9148 | 0.9390 |
| | Enhanced Representation | 0.7096 | 0.8832 | 0.3537 | 0.8901 | 0.7836 | 0.8377 | 0.8749 | 0.8330 | 0.8831 |

* $m$ (embedding dimension) and $\tau$ (time-lag) are two parameters used for calculating PE, and the values are selected following [30, 16].

As can be observed, the enhanced representations learnt by SCINet indeed have lower PE values compared with the original inputs, which indicates that it is easier to predict the future from the enhanced representations using the same forecaster.

### 4.3 Ablation studies

To evaluate the impact of each main component used in SCINet, we experiment on several model variants on two datasets: *ETTh1* and *PEMS08*.

**SCIBlock**: we first set the number of stacks $K = 1$ and the number of SCINet levels $L = 3$. For the SCI-Block design, to validate the effectiveness of the interactive learning and the distinct convolution weights for processing the sub-sequences, we experiment on two variants, namely *w/o. InterLearn* and *WeightShare*. The *w/o. InterLearn* is obtained by removing the interactive-learning procedure described in Eq. (1) and (11). In this case, the two sub-sequences would be updated using $\mathbf{F}'_{odd} = \rho(\phi(\mathbf{F}_{odd}))$ and $\mathbf{F}'_{even} = \eta(\psi(\mathbf{F}_{even}))$. For *WeightShare*, the modules $\phi$, $\rho$, $\psi$, and $\eta$ share the same weight.

The evaluation results in Fig. 4 show that both interactive learning and distinct weights are essential, as they improve the prediction accuracies of both datasets at various prediction horizons. At the same time, comparing Fig. 4(a) with Fig. 4(b), we can observe that interactive learning is more effective for cases with longer look-back window sizes. This is because, intuitively, we can extract more effective features by exchanging information between the downsampled sub-sequences when there are longer look-back windows for such interactions.

**SCINet**: for the design of SCINet with multiple levels of SCI-Blocks, we also experiment on two variants. The first variant *w/o. ResConn* is obtained by removing the residual connection from the complete SCINet. The other variant *w/o. Linear* removes the decoder (i.e., the fully-connected layer) from the complete model. As can be observed in Fig. 4, removing the residual connection leads to a significant performance drop. Besides the general benefit in facilitating the model training, more importantly, the predictability of the original time series is enhanced with the help of residuals. The fully-connected layer is also critical for prediction accuracy, indicating the effectiveness of

---

[2]Please note that PE is only a quantitative measurement based on complexity. It would not be proper to say that a time series with lower PE value will be always easier to predict than a different type of time series with a higher PE value because the prediction accuracy also depends on many other factors, such as the available data for training, the trend and seasonality elements of the time series data, as well as the predictive model.

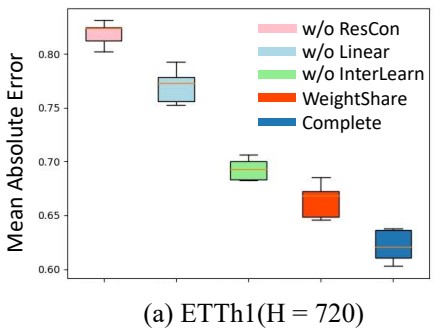 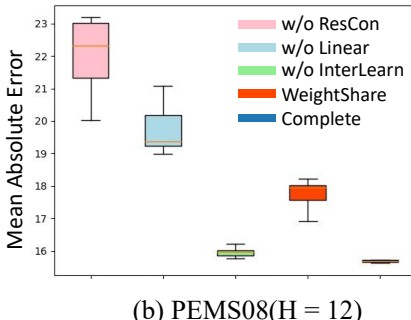

(a) ETTh1(H = 720)          (b) PEMS08(H = 12)

Figure 4: Component analysis of SCINet on two datasets. Smaller values are better. See Section 4.3.

the decoder in extracting and fusing the most relevant temporal information according to the given supervision for prediction.

We also conduct comprehensive ablation studies on the impact of $K$ (number of stacks) and $L$ (number of levels), and the selection of operator in the interact learning mechanism. These results are shown in the Appendix B.2 due to space limitation.

## 5 Limitations and Future Work

In this paper, we mainly focus on TSF problem for the *regular time series* collected at even intervals of time and ordered chronologically. However, in real-world applications, the time series might contain noisy data, missing data or collected at irregular time intervals, which is referred to as *irregular time series*. The proposed SCINet is relatively robust to the noisy data thanks to the progressive downsampling and interactive learning procedure, but it might be affected by the missing data if the ratio exceeds a certain threshold, wherein the downsampling-based multi-resolution sequence representation in SCINet may introduce biases, leading to poor prediction performance. The proposed downsampling mechanism may also have difficulty handling data collected at irregular intervals. We plan to take the above issues into consideration in the future development of SCINet.

Moreover, this work focuses on the deterministic time series forecasting problem. Many application scenarios require probabilistic forecasts, and we plan to revise SCINet to generate such prediction results.

Finally, while SCINet generates promising results for spatial-temporal time series without explicitly modeling spatial relations, the forecasting accuracy could be further improved by incorporating dedicated spatial models. We plan to investigate such solutions in our future work.

## 6 Conclusion

In this paper, we propose a novel neural network architecture, sample convolution and interaction network (*SCINet*) for time series modeling and forecasting, motivated by the unique properties of time series data compared to generic sequence data. The proposed SCINet is a hierarchical downsample-convolve-interact structure with a rich set of convolutional filters. It iteratively extracts and exchanges information at different temporal resolutions and learns an effective representation with enhanced predictability. Extensive experiments on various real-world TSF datasets demonstrate the superiority of our model over state-of-the-art methods.

## Acknowledgments and Disclosure of Funding

This work was supported in part by Alibaba Group Holding Ltd. under Grant No. TA2015393. We thank the anonymous reviewers for their constructive comments and suggestions.

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
