# Appendix

In this appendix, we first introduce the datasets and evaluation metrics used in the experiments in Section A. Then, we provide extra experimental results in Section B. In Section C, we present details of network design, training scheme, and hyper-parameter tuning.

## A    Datasets and Evaluation Metrics

We conduct experiments on 11 popular time series datasets: (1) *Electricity Transformer Temperature* [42] (ETTh(1,2),ETTm1) [3] consists of 2 year electric power data collected from two separated counties of China. Each data point includes an "oil temperature" value and 6 power load features. (2) *Traffic* [4] contains the hourly data describing the road occupancy rates (ranging from 0 to 1) that are recorded by the sensors on San Francisco Bay area freeways from 2015 to 2016 (48 months in total). (3) *Solar-Energy* [5] records the solar power production from 137 PV plants in Alabama State, which are sampled every 10 minutes in 2016. (4) *Electricity* [6] includes the hourly electricity consumption (kWh) records of 321 clients from 2012 to 2014. (5) *Exchange-Rate* [7] collects the daily exchange rates of 8 foreign countries from 1990 to 2016. (6) *PeMS* [8] contains four public traffic network datasets (*PEMS03, PEMS04, PEMS07 and PEMS08*) which are respectively constructed from Caltrans Performance Measurement System (PeMS) of four districts in California. The data is aggregated into 5-minutes windows, resulting in 12 points per hour and 288 points per day.

### A.1    Electricity Transformer Temperature (ETT)

For data pre-processing, we perform zero-mean normalization, i.e., $X' = (X - mean(X))/std(X)$, where $mean(X)$ and $std(X)$ are the mean and the standard deviation of historical time series, respectively. We use Mean Absolute Errors (MAE) [17] and Mean Squared Errors (MSE) [26] for model comparison. Besides, the train, validation and test sets contain 12, 4 and 4 months data, respectively.

$$MAE = \frac{1}{\tau} \sum_{i=0}^{\tau} |\hat{x}_i - x_i| \tag{5}$$

$$MSE = \frac{1}{\tau} \sum_{i=0}^{\tau} (\hat{x}_i - x_i)^2 \tag{6}$$

where $\hat{x}_i$ is the model's prediction, and $x_i$ is the ground-truth. $\tau$ is the length of the prediction horizon.

### A.2    PeMS

Following [17], the data is pre-processed using zero-mean normalization and we use Root Mean Squared Errors (RMSE) and Mean Absolute Percentage Errors (MAPE) as evaluation metrics on this dataset.

$$RMSE = \sqrt{\frac{1}{\tau} \sum_{i=0}^{\tau} (\hat{x}_i - x_i)^2}, \tag{7}$$

$$MAPE = \sqrt{\frac{1}{\tau} \sum_{i=0}^{\tau} |(\hat{x}_i - x_i)/x_i|}. \tag{8}$$

### A.3    Traffic, Solar-Energy, Electricity and Exchange-Rate

In our experiments, the length of the look-back window $T$ for these datasets is 168, and we trained independent models for different length of future horizon (i.e., $\tau = 3, 6, 12, 24$). We use Root Relative

---

[3] https://github.com/zhouhaoyi/ETDataset

[4] http://pems.dot.ca.gov

[5] http://www.nrel.gov/grid/solar-power-data.html

[6] https://archive.ics.uci.edu/ml/datasets/ElectricityLoadDiagrams20112014

[7] https://github.com/laiguokun/multivariate-time-series-data

[8] https://pems.dot.ca.gov

Squared Error (RSE) and Empirical Correlation Coefficient (CORR) to evaluate the performance of the TSF models on these datasets following [19], which are calculated as follows:

$$RSE = \frac{\sqrt{\sum_{i=0}^{\tau}(\hat{x}_i - x_i)^2}}{\sqrt{\sum_{i=0}^{\tau}(x_i - mean(X))^2}}, \tag{9}$$

$$CORR = \frac{1}{d}\sum_{j=0}^{d}\frac{\sum_{i=0}^{\tau}(x_{i,j} - mean(X_j))(\hat{x}_{i,j} - mean(\hat{X}_j))}{\sum_{i=0}^{\tau}(x_{i,j} - mean(X_j))^2(\hat{x}_{i,j} - mean(\hat{X}_j))^2}, \tag{10}$$

where $X$ and $\hat{X}$ are the ground-truth and model's prediction, respectively. $d$ is the number of variates.

## B  Extra Experimental Results

In this section, we first add error bars on different forecasting steps T, and also conduct empirical studies on *ETTh1* and *PEMS* datasets to show the impact of different parameter and operator combinations in *SCI-Block*.

### B.1  Error Bars Evaluation

Since deep models for time series forecasting may be influenced by different random initialization, we report our results with 5 runs on the ETTh1 dataset. From Table 8, we show the standard deviation (Std.) is basically 2% to 3% of the mean values, indicating SCINet is robust towards different initialization.

Table 8: The error bars of SCINet with 5 runs on the ETTh1 dataset.

| T | Metrics | Seed 1 | Seed 2 | Seed 3 | Seed 4 | Seed 5 | Mean | Std. |
|---|---|---|---|---|---|---|---|---|
| 24 | MSE | 0.3346 | 0.3381 | 0.3541 | 0.3370 | 0.3370 | 0.3402 | 0.0079 |
| | MAE | 0.3699 | 0.3742 | 0.3826 | 0.3719 | 0.3722 | 0.3742 | 0.0050 |
| 48 | MSE | 0.4148 | 0.4259 | 0.3899 | 0.3830 | 0.3856 | 0.3998 | 0.0193 |
| | MAE | 0.4370 | 0.4520 | 0.4139 | 0.4108 | 0.4173 | 0.4262 | 0.0177 |
| 168 | MSE | 0.4490 | 0.5038 | 0.4433 | 0.4493 | 0.4432 | 0.4577 | 0.0259 |
| | MAE | 0.4526 | 0.4985 | 0.4466 | 0.4501 | 0.4476 | 0.4591 | 0.0222 |
| 336 | MSE | 0.5288 | 0.5935 | 0.5230 | 0.5308 | 0.5373 | 0.5427 | 0.0289 |
| | MAE | 0.5131 | 0.5486 | 0.5114 | 0.5150 | 0.5166 | 0.5209 | 0.0156 |
| 720 | MSE | 0.5607 | 0.5923 | 0.5855 | 0.5582 | 0.5678 | 0.5729 | 0.0152 |
| | MAE | 0.5469 | 0.5653 | 0.5630 | 0.5418 | 0.5502 | 0.5534 | 0.0103 |

### B.2  Evaluation on the Impact of $K$ and $L$

We conduct experiments on *ETTh1* dataset (with the multivariate experimental setting) to evaluate the impact of $K$ (number of stacks) and $L$ (number of levels), under various look-back window sizes $T$. The prediction horizon is fixed to be 24.

As can be observed from Table 9, when fixing $K = 1$, larger $L$ leads to better prediction accuracy for the cases with larger $T$ ($T = 128$ or $192$). This is because we could further extract essential information from coarser temporal resolutions with deeper levels in the SCINet when $T$ is large. As for the number of stacks $K$, when fixing $L = 3$, if $T$ is small (e.g. $T = 24$ or $48$), we find that increasing $K$ would improve prediction accuracy. This is because, under such circumstances, the information extracted from a single SCINet is insufficient. By stacking more SCINets, we effectively increase the representation learning capability of the model, which facilitates extracting more robust temporal relations for the forecasting task. However, when $T$ is large (e.g., $192$), a shallow stack can already well capture the temporal dependencies for the time series. Under such circumstances, using deeper stacks may suffer from overfitting issues with the increase of parameters, which degrades the performance in the inference stage.

From Table 9, we can observe a clear trade-off between $L$ and $K$. Moreover, the performance variation under different $T$ also indicates the importance of the look-back window selection for forecasting tasks. While $T$ is typically pre-determined based on domain knowledge about the time

Table 9: The impact of $L$ and $K$ on MSE.

| Number of | Horizon | 24 | | | | |
|---|---|---|---|---|---|---|
| Levels & Stacks | $T$ | 24 | 48 | 96 | 128 | 192 |
| Level $L$ ($K$ =1) | 2 | 0.411 | 0.348 | 0.347 | 0.334 | 0.384 |
| | 3 | **0.405** | **0.346** | **0.316** | 0.418 | 0.330 |
| | 4 | - | 0.360 | 0.340 | 0.331 | **0.325** |
| | 5 | - | - | 0.354 | **0.323** | 0.356 |
| Stack $K$ ($L = 3$) | 1 | 0.405 | 0.346 | **0.316** | 0.418 | **0.330** |
| | 2 | 0.423 | 0.344 | 0.344 | **0.339** | 0.375 |
| | 3 | **0.374** | **0.341** | 0.345 | 0.353 | 0.363 |
| | 4 | 0.390 | 0.342 | 0.335 | 0.356 | 0.388 |

- Dash denotes the input cannot be further splitted.

series data, based on our empirical study, $L \leq 5$ and $K \leq 3$ are usually sufficient and tuning these hyperparameters does not incur much effort.

### B.3 Empirical Study on Operator Selection

In interactive-learning equation,

$$\mathbf{F}'_{odd} = \mathbf{F}^s_{odd} \pm \rho(\mathbf{F}^s_{even}), \quad \mathbf{F}'_{even} = \mathbf{F}^s_{even} \pm \eta(\mathbf{F}^s_{odd}). \tag{11}$$

the operators can be either "addition" or "subtraction". Although the model can learn the operation adaptively during training, the parameter initialization would affect the final performance. As shown in the following table the impact of operator settings is minor.

Table 10: The impact of different operators

| Operators | PEMS03 | PEMS04 | PEMS07 | PEMS08 |
|---|---|---|---|---|
| | MAE | | | |
| +, + | 15.08 | 19.27 | 21.69 | **15.72** |
| -, - | **15.06** | **19.21** | **21.63** | 15.78 |
| +, - | 15.09 | 19.31 | 21.77 | 15.84 |
| -, + | 15.30 | 19.32 | 21.72 | 15.79 |

## C Reproducibility

Our code is implemented with PyTorch. All the experiments are conducted on an Nvidia Tesla V100 SXM2 GPU (32GB memory), which is sufficient for all our experiments.

**Structure of the network modules $\phi$, $\rho$, $\psi$, and $\eta$ in SCI-Block:** As shown in Fig. 5, $\phi$, $\rho$, $\psi$, and $\eta$ use the same network architecture. First, the replication padding is used to keep the border shrunk caused by the convolution operation. Then, a 1d convolutional layer with kernel size $k$ is applied to extend the input channel $C$ to $h*C$ and followed with LeakyRelu and Dropout. $h$ means a scale of the hidden size. Next, the second 1d convolutional layer with kernel size $k$ is to recover the channel $h*C$ to the input channel $C$. The stride of all the convolutions is 1. We use a LeakyRelu activation after the first convolutional layer because of its sparsity properties and a reduced likelihood of vanishing gradient. We apply a Tanh activation after the second convolutional layer since it can keep both positive and negative features into [-1, 1].

**Loss Function**

To enhance the performance in single-step (short-term time series forecasting Sec. 4.2) forecasting, we revise the loss function of the last SCINet in the stacked SCINet with $K(K \geq 1)$. The loss function contains two parts:

$$\mathcal{L}_k = \frac{1}{\tau} \sum_{i=0}^{\tau} \left\| \hat{\mathbf{x}}_i^k - \mathbf{x}_i \right\|, \quad k \neq K. \tag{12}$$

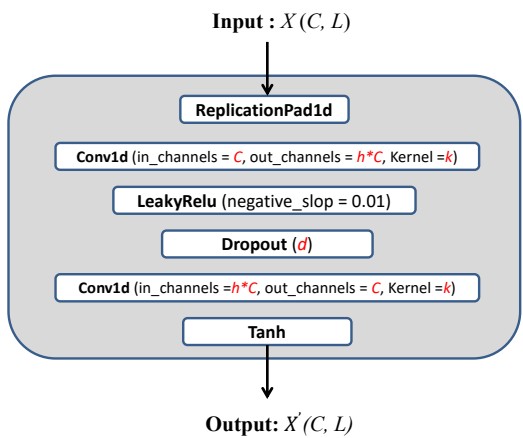

**Input :** $X(C, L)$

ReplicationPad1d

Conv1d (in_channels = $C$, out_channels = $h*C$, Kernel = $k$)

LeakyRelu (negative_slop = 0.01)

Dropout ($d$)

Conv1d (in_channels = $h*C$, out_channels = $C$, Kernel = $k$)

Tanh

**Output:** $X'(C, L)$

Figure 5: The structure of $\phi$, $\rho$, $\psi$, and $\eta$.

For the last stack $K$, we introduce a balancing parameter $\lambda \in (0, 1)$ for the value of the last time-step[9]:

$$\mathcal{L}_K = \frac{1}{\tau - 1} \sum_{i=0}^{\tau-1} \left\| \hat{\mathbf{x}}_i^K - \mathbf{x}_i \right\| + \lambda \left\| \hat{\mathbf{x}}_\tau^K - \mathbf{x}_\tau \right\|. \tag{13}$$

Therefore, the total loss of the stacked SCINet can be written as:

$$\mathcal{L} = \sum_{k=1}^{K-1} \mathcal{L}_k + \mathcal{L}_K. \tag{14}$$

**Training details:** For all datasets, we fix the random seed to be 4321, and train the model for 150 epochs at most. The reported results on the test set are based on the model that achieves the best performance on the validation set.

**Hyper-parameter tuning:** We conduct a grid search over all the essential hyper-parameters on the held-out validation set of the datasets. The detailed hyper-parameter configurations of *ETT* are shown in Table 11 [10]. Besides, the parameters of the four datasets in *PeMS* are presented in Table13. The *Traffic, Solar-Energy, Electricity and Exchange-rate* are shown in Table 12. Notably, we only apply the weighted loss to the *Solar* and *Exchange-rate* data since they show less auto-correlation [19], which indicates the temporal correlation of the distant time-stamp cannot be well modelled by a general L1 loss. Moreover, to build a non-causal TCN[11] in the paper, we only need to remove the *chomps* in the code and make the padding equal to the dilation.

---

[9]This is slightly different from other practice for single-step forecasting [19], because we choose to use all the available values in the prediction window as supervision signal.

[10]The results on ETTh2 and ETTm1 datasets can be referred to: `https://github.com/cure-lab/SCINet`

[11]https://github.com/locuslab/TCN/issues/45

Table 11: The hyperparameters in ETT datasets (Multivariate)

| Model configurations | | ETTh1 | | | | | ETTh2 | | | | | ETTm1 | | | | |
|---|---|---|---|---|---|---|---|---|---|---|---|---|---|---|---|---|
| Hyperparameter | Horizon | 24 | 48 | 168 | 336 | 720 | 24 | 48 | 168 | 336 | 720 | 24 | 48 | 96 | 288 | 672 |
| | Look-back window | 48 | 96 | 336 | 336 | 736 | 48 | 96 | 336 | 336 | 736 | 48 | 96 | 384 | 672 | 672 |
| | Batch size | 8 | 16 | 32 | 512 | 256 | 16 | 4 | 16 | 128 | 128 | 32 | 16 | 32 | 32 | 32 |
| | Learning rate | 3e-3 | 9e-3 | 5e-4 | 1e-4 | 5e-5 | 7e-3 | 7e-3 | 5e-5 | 5e-5 | 1e-5 | 5e-3 | 1e-3 | 5e-5 | 1e-5 | 1e-5 |
| SCI Block | h | 4 | 4 | 4 | 1 | 1 | 8 | 4 | 0.5 | 1 | 4 | 4 | 4 | 0.5 | 4 | 4 |
| | k | 5 | 5 | 5 | 5 | 5 | 5 | 5 | 5 | 5 | 5 | 5 | 5 | 5 | 5 | 5 |
| | Dropout | 0.5 | 0.25 | 0.5 | 0.5 | 0.5 | 0.25 | 0.5 | 0.5 | 0.5 | 0.5 | 0.5 | 0.5 | 0.5 | 0.5 | 0.5 |
| SCINet | L (level) | 3 | 3 | 3 | 4 | 5 | 3 | 4 | 4 | 4 | 5 | 3 | 4 | 4 | 5 | 5 |
| Stacked SCINet | K (stack) | 1 | 1 | 1 | 1 | 1 | 1 | 1 | 1 | 1 | 1 | 1 | 2 | 2 | 1 | 2 |

Table 12: The hyperparameters in Traffic, Solar-energy, Electricity and Exchange-rate datasets

| Model configurations | | Solar | | | | Electricity | | | | Traffic | | | | Exc-Rate | | | |
|---|---|---|---|---|---|---|---|---|---|---|---|---|---|---|---|---|---|
| Hyperparameter | Horizon | 3 | 6 | 12 | 24 | 3 | 6 | 12 | 24 | 3 | 6 | 12 | 24 | 3 | 6 | 12 | 24 |
| | Look-back window | 160 | | | | 168 | | | | | | | | | | | |
| | Batch size | 256 | 256 | 1024 | 256 | 32 | | | | 16 | | | | 4 | | | |
| | Learning rate | 1e-4 | | | | 9e-3 | | | | 5e-4 | | | | 5e-3 | | | 7e-3 |
| SCI Block | h | 1 | 0.5 | 2 | 1 | 8 | | | | 1 | 2 | 0.5 | 2 | 0.125 | | | |
| | k | 5 | | | | 5 | | | | 5 | | | | 5 | | | |
| | Dropout | 0.25 | | | | 0 | | | | 0.5 | 0.25 | 0.25 | 0.5 | 0.5 | | | |
| SCINet | L (level) | 4 | | | | 3 | | | | 3 | | | 2 | 3 | | | |
| Stacked SCINet | K (stack) | 2 | | | 1 | 2 | | | | 2 | 1 | 2 | 2 | 1 | | | |
| | Loss weight ($\lambda$) | 0.5 | | | | × | | | | × | | | | 0.5 | | | |

Table 13: The hyperparameters in PeMS datasets

| Model configurations | | PEMS03 | PEMS04 | PEMS07 | PEMS08 |
|---|---|---|---|---|---|
| Hyperparameter | Horizon | 12 | | | |
| | Look-back window | 12 | | | |
| | Batch size | 8 | | | |
| | Learning rate | 1e-3 | | | |
| SCI Block | h | 0.0625 | 0.0625 | 0.03125 | 1 |
| | k | 5 | | | |
| | Dropout | 0.25 | 0 | 0.25 | 0.5 |
| SCINet | L (level) | 2 | | | |
| Stacked SCINet | K (stack) | 1 | | | |