# OpenReview forum: "SCINet: Time Series Modeling and Forecasting with Sample Convolution and Interaction"
_NeurIPS.cc/2022/Conference — NeurIPS 2022 Accept_

### Official Review · Reviewer_pviL · 2022-06-20

**Rating:** 5
**Confidence:** 3
**Soundness:** 3 good
**Presentation:** 2 fair
**Contribution:** 2 fair

**Summary:**

The article proposes a novel method based on hierarchical downsample-convolve-interact architecture, named SCINet. The article points out that the temporal relation of time-series data is maintained even after downsampling, and argues that multiple convolution filters extract rich features from downsampled sub-sequences. The SCI block, the basic unit of SCINet, utilizes different convolution kernels, an inter-learning approach is applied, has residual connections, and linear enhancement is applied. The proposed method was evaluated on 9 types of popular time-series dataset and compared with the latest methods. Performance degradation occurred in the ETTh1 [37] dataset in the long-term forecasting task, but a significant improvement in overall performance was verified.

**Questions:**

1. Is the improved performance achieved by stacking SCI-blocks, or is it due to the residual connection between SCI-blocks, or is it due to the SCI-block itself? Considering that the comparative methods also handle downsampled features by multiple convolution, it is necessary to highlight the cause of the performance improvement.

**Limitations:**

The authors point out that if the time-series obtained in the real-world are noisy or greatly affected by missing data, the method can lead to degradation in performance.

**Strengths And Weaknesses:**

(strength)
The proposed method has achieved improved performance compared to the latest deep learning-based time-series modeling methods. The proposed method is technically sound and has a sufficient amount of validation.
(weakness)
The article mixes several ideas, including SCI-block, hierarchical structure, downsampling, and residual connection.

---

> ### Author Response · Authors · 2022-08-02
> **Response to Reviewer pviL**
>
> **Q1: The article mixes several ideas, including SCI-block, hierarchical structure, downsampling, and residual connection. The idea may seem like lego-style work, in that the pytorch/tensorflow library provides a ready-to-go functions.**
>
> A: We **do NOT agree** with the reviewer on saying "the idea may seem like lego-style work". To the best of our knowledge, SCINet is the first of its kind that conducts sample convolution and interaction for time series analysis, and "pytorch/tensorflow library **certainly does NOT** provide a ready-to-go function.
>
> Compared to the existing temporal convolutional network (TCN) that extracts the average temporal features from the data/features in the previous layer of the network with the shared convolutional filter, SCINet employs multiple convolutional filters at every layer of the hierarchical network, facilitating the effective extraction of temporal dynamics. This is not possible without sampling sub-sequences first. We have also designed an interactive learning module to allow information interchange between the downsampled sub-sequences. All the above concepts are new for time series modeling and forecasting. Experimental results show that SCINet achieves significant forecasting accuracy improvements over both TCNs and Transformer-based solutions.
>
> **Q2: Is the improved performance achieved by stacking SCI-blocks, or is it due to the residual connection between SCI-blocks, or is it due to the SCI-block itself? Considering that the comparative methods also handle downsampled features by multiple convolution, it is necessary to highlight the cause of the performance improvement.**
>
> A: The main contributor to the significant performance benefits of SCINet is the SCI-Block design. As can be seen in Tables 4-6 in the supplementary material, the best result is obtained with only one stack in most cases. Residual connection is essential, but the residual connection in our design is simply connecting the original time series information in the look-back window, which is different from the intermediate residual connection used in other networks such as ResNet.
>
> To the best of our knowledge, **we are not aware of any other comparative methods that handle dowsampled features by multiple convolution in the literature.** We would appreciate the reviewer to point out the specific work so that we can examine the differences.

---

> > ### Comment · Reviewer_pviL · 2022-08-03
> > **I would like to reconsider the contribution (1 poor) of the original review.**
> >
> > 1. The authors addressed that the TCN structure processes downsampled features through a single shared convolution filter and argued that the proposed method aggregates global/local views because it uses multiple convolutions (line 91- 97 in page 3, Line 138-143 in page 4, and line 175-179 in page 5). I realized the proposed method's difference through the authors' rebuttals, and I would like to reconsider the contribution (1 poor) of the original review.
> >
> > 2. I again reviewed the source code of SCI-block on the authors' anonymous Github. I have concluded that the authors claim that multiple convolutions of downsampled features (consisting of separate weights) cannot be implemented with lego-style work or ready-to-go functions.
> >
> > 3. But instead, considering that the concept of introducing residual connection and processing downsampled features individually is not new, it seems that the current presentation should be reinforced (since several ideas are mentioned, SCI-Net's technical novelty may seem obscure).
> > So far, I understood that the main contributor to the performance enhancement is the design of SCI-Block, that the convolutional layer processes the downsampled features separately and aggregates essential information extracted from two subsequences. If so, I think it would be helpful to highlight an interactive-learning strategy (lines 121-126). For example, how about adding an algorithm or flowchart of an interactive-learning strategy?

---

> > > ### Author Response · Authors · 2022-08-03
> > > **Thank you for the confirmation of the novelty of our work!**
> > >
> > > We appreciate the reviewer's confirmation of the novelty of our work. We address the new comments in the following.
> > >
> > > Q1: **But instead, considering that the concept of introducing residual connection and processing downsampled features individually is not new, it seems that the current presentation should be reinforced (since several ideas are mentioned, SCI-Net's technical novelty may seem obscure).**
> > >
> > > A: As shown in the Introduction section, we list the two main contributions of this work as follows:
> > > 1. We propose SCINet, a hierarchical downsample-convolve-interact TSF framework that effectively models time series with complex temporal dynamics ...
> > > 2. We design the basic building block, SCI-Block, for constructing SCINet, which downsamples the input data/feature into two sub-sequences, and then extracts features of each sub37 sequence using distinct convolutional filters. To compensate for the information loss during the downsampling procedure, we incorporate interactive learning between the two convolutional features within each SCI-Block.
> > >
> > > We do not consider residual connection and stacked network as the contributions of this work. We shall further clarify this issue in the main body of the paper in the revised version.
> > >
> > > Q2: **So far, I understood that the main contributor to the performance enhancement is the design of SCI-Block, that the convolutional layer processes the downsampled features separately and aggregates essential information extracted from two subsequences. If so, I think it would be helpful to highlight an interactive-learning strategy (lines 121-126). For example, how about adding an algorithm or flowchart of an interactive-learning strategy?**
> > >
> > > A: In fact, Lines 127-137 detail the interactive-learning strategy, and it can be also observed in Fig. 2(a). Further details can be found in our code. At the same time, we would like to emphasize that the main contributor to the performance of SCINet is the sample convolution strategy, which enables the effective extraction of temporal dynamics. In many cases, the forecasting accuracy is still quite competitive without interactive learning.

---

### Official Review · Reviewer_B5XJ · 2022-07-03

**Rating:** 5
**Confidence:** 4
**Soundness:** 3 good
**Presentation:** 4 excellent
**Contribution:** 3 good

**Summary:**

This paper devises a new architecture, SCINet, for the time-series forecasting problem. To extract features, SCINet mainly adopts convolutions on sub-series with different resolutions, which is accompanied by additional operations to ensure the interactions among the subsequences. The proposed architecture can also be stacked to improve forecasting accuracy. SCINet is compared with other convolutional models and transformer-based models on short-term/long-term/spatial-temporal time series forecasting tasks.

**Questions:**

1. In line 109, the authors state that “each SCI-Block has both local and global views of the entire time series”. I would appreciate it if the authors could elaborate more on how a single SCI-Block can attain the global view of entire time series.
2. I would also appreciate it if the authors could explain more about the claim – “the proposed downsample-convolve-interact architecture achieves an even larger receptive field at each convolutional layer” in line 139.
3. As SCI-Net is claimed to “capture both short-term and long-term temporal dependencies in the time series”, I think it would be helpful to show its superiority by comparing it with Transformer-XL (which is good at preserving long-term information) empirically.
4. Besides, as SCI-Net and Pyraformer both adopt hierarchical architecture for time-series forecasting task, I think it’s worth adding the latter in the experiment section for comparison.
5. In the Table 3&4 of the Supplementary Material, the authors report the hyperparameters used by the SCI-Net. I notice that the batch size and learning rate significantly vary both within a single dataset / across the datasets. I’m wondering whether the performance of SCI-Net is highly sensitive to these training strategies.
6. In Table 3 of the main text, I notice that the paper uses the performance reported in the Autoformer and Informer paper. However, Autoformer and Informer adopt a significantly different training strategies (batch size, learning rate, and training epochs) compared with the SCI-Net. More concretely, Autoformer and Informer use constant batch size and same initial learning rate, respectively; while SCI-Net tunes these hyperparameters for every setting. Moreover, Autoformer and Informer are only trained for 10 and 8 epochs, respectively; while SCI-Net is trained for 150 epochs. Therefore, I’m wondering whether Table 3 offers a fair comparison and whether SCI-Net is efficient in terms of training time and generalizable to different datasets without intensive hyperparameter tuning.
7. It would be helpful if the authors could provide how many times each experiment is repeated and the error bar (e.g., standard deviation) of the reported results.


**Limitations:**

The limitations have been covered properly by the authors in the corresponding section.

**Strengths And Weaknesses:**

Strengths:
1. The paper is well written and easy to follow.

2. Though convolution is not uncommon in TSF models, the paper utilizes the special properties of time series data and proposes the downsampling and interaction operations, which leads to good empirical performance.

Weaknesses:
1. More empirical comparison with other related models would be better to support the superiority of the SCI-Net. (Please refer to Questions 3-4 for details)

2. More explanations on the training strategies adopted would be helpful to prove that the experiment section gives a fair comparison with other works. (Please refer to Questions 5-7 for details)

---

> ### Author Response · Authors · 2022-08-02
> **Response to Reviewer B5XJ**
>
> **Q1: In line 109, the authors state that “each SCI-Block has both local and global views of the entire time series”. I would appreciate it if the authors could elaborate more on how a single SCI-Block can attain the global view of entire time series.**
>
> A: The separate convolutional filters for each of the two sub-sequences in an SCI-Block extract the local features, and the interactive learning module exchanges information between all the features extracted from the two sub-sequences, thereby having a global view of the entire series.
>
> **Q2: I would also appreciate it if the authors could explain more about the claim – “the proposed downsample-convolve-interact architecture achieves an even larger receptive field at each convolutional layer” in line 139.**
>
> A: Consider a convolutional filter with kernel size $k$, for the bottom layer of the conventional temporal convolutional network (TCN), the receptive field of each convolutional filter is simply $k$. However, for SCINet, due to downsampling, the receptive field of each convolutional filter is already enlarged to roughly $2k$. Moreover, the interactive learning module enables the reception of features from the other branch in the SCI-Block, thereby further enlarging the receptive field.
>
> **Q3: As SCI-Net is claimed to “capture both short-term and long-term temporal dependencies in the time series”, I think it would be helpful to show its superiority by comparing it with Transformer-XL ...**
>
> A: We thank the reviewer for the suggestion. Transformer-XL is an Attentive Language Models Beyond a Fixed-Length Context, and it has not been used for time series analysis. At the same time, there are several Transformer-based models (e.g., Informer (AAAI2021), Autoformer  (NeuIPS 2021), and Pyraformer (ICLR 2022)) for long-term time series forecasting, which claim to preserve long-term information. We have compared with these works in our experimental results, and SCINet can surpass them by a large margin, as shown in Table 3.
>
> **Q4: Besides, as SCI-Net and Pyraformer both adopt hierarchical architecture for time-series forecasting task, I think it’s worth adding the latter in the experiment section for comparison.**
>
> A: We thank the reviewer for pointing out this related work. We compare SCINet with Pyraformer under different settings in the following table.
> |       |          |     Pyraformer |    | SCINet |        |   |   |   |   |
> |-------|----------|--------|------------|-----------|-----------|---|------|---|---|
> |       | Pred_len | MSE        | MAE   | MSE    | MAE    |   |   |   |   |
> | ETTh1 | 192      | 0.808      | 0.683 | 0.451  | 0.457  |   |   |   |   |
> |       | 336      | 0.945      | 0.766 | 0.502  | 0.497  |   |   |   |   |
> |       | 720      | 1.022      | 0.806 | 0.583  | 0.56   |   |   |   |   |
> | ETTm1 | 96       | 0.48       | 0.486 | 0.191  | 0.365  |   |   |   |   |
> |       | 288      | 0.754      | 0.659 | 0.365  | 0.415  |   |   |   |   |
> |       | 672      | 0.857      | 0.707 | 0.713  | 0.604  |   |   |   |   |
>
> As can be observed from the table, SCINet can surpass Pyraformer by a large margin.
>
> **Q5: In the Table 3&4 of the Supplementary Material, ..., the batch size and learning rate significantly vary both within a single dataset / across the datasets. I’m wondering whether the performance of SCI-Net is highly sensitive to these training strategies.**
>
> A: Time series datasets vary significantly as they are from various domains (e.g., finance, energy, and traffic), which affects the best hyperparameters used in training across different datasets. Based on our experiments, if we use the same hyperparameter setting under different forecasting horizons for the same dataset, the performance of SCINet does not change much.
>
> **Q6: In Table 3 of the main text, ...,  Autoformer and Informer adopt a significantly different training strategies (batch size, learning rate, and training epochs) compared with the SCI-Net ... Therefore, I’m wondering whether Table 3 offers a fair comparison and whether SCI-Net is efficient in terms of training time and generalizable to different datasets without intensive hyperparameter tuning.**
>
> A: Indeed, we have reported the best results with hyperparameter tuning. This is common practice and our model is not highly sensitive to hyperparameters. We have tried to train Autofomer and Informer for more epochs and their performances do not improve. We have not tried different batch sizes and learning rates due to the short rebuttal time constraints. Also, considering generating lots of models for various datasets is not a practical requirement, we do not see tuning as a practical constraint. Please also refer to our answer to Q5.
>
> **Q7: It would be helpful if the authors could provide ...  error bar (e.g., standard deviation) of the reported results.**
>
> We thank the reviewer for the suggestion and we have added the error bars for ETTh1 dataset to the revised supplementary materials (see Table 1).

---

> > ### Comment · Reviewer_B5XJ · 2022-08-04
> > **Thank you for the response.**
> >
> > I appreciate that the authors have cleared up my doubts / confusions related to Q1-Q4 and Q7. However, the authors' responses did not fully address my concerns over the training efficiency, including training time and intensive hyper-parameter tuning.
> > * I have checked the methods (Autoformer, Informer, Pyraformer and Reformer) that SCINet was compared to in the paper, none of them adopt such diverse hyper-parameter settings (batch size and learning rate) for a single dataset. Even for some other recent time-series models (FEDformer and FiLM), which have comparable performances with the SCINet, their training strategies are also much more consistent. Therefore, I would appreciate it if the authors could further provide more evidence that "Based on our experiments, if we use the same hyperparameter setting under different forecasting horizons for the same dataset, the performance of SCINet does not change much.". Otherwise, I'm concerned with the model's generalizability in the real applications.
> >
> > * Again, the models I mentioned in the first point are usually trained with fewer than 20 epochs (while SCINet using150 epochs). I believe an additional comparison on the training and inference speed will be helpful to support the SCINet's efficiency.

---

> > > ### Author Response · Authors · 2022-08-06
> > > **Clarification on Training Efficiency**
> > >
> > > **Q1: I have checked the methods (Autoformer, Informer, Pyraformer and Reformer) that SCINet was compared to in the paper, none of them adopt such diverse hyper-parameter settings (batch size and learning rate) for a single dataset. Even for some other recent time-series models (FEDformer and FiLM), which have comparable performances with the SCINet, their training strategies are also much more consistent. Therefore, I would appreciate it if the authors could further provide more evidence that "Based on our experiments, if we use the same hyperparameter setting under different forecasting horizons for the same dataset, the performance of SCINet does not change much.". Otherwise, I'm concerned with the model's generalizability in the real applications.**
> > >
> > > A: As can be seen in Table 5 of the supplementary material, for the 4 datasets (Solar, Electricity, Traffic, and Exchange Rate), the hyper-parameter settings (batch size and learning rate) for the same dataset with various forecasting horizons are almost the same (14 out of 16 cases). As shown in Table 6, for the PEMS datasets (PEMS03, PEMS04, PEMS07, and PEMS08), we have used the same hyper-parameter settings across these clolsely related datasets. Therefore, the diverse hyper-parameter settings only manifest themselves on the ETT datasets. We attribute it to the fact that ETT datasets are small-sized and noisy compared to other datasets, which may easily overfit the model without properly tuning the hyper-parameters.
> > >
> > > At the same time, we would like to point out that all the mentioned Transfomer-based solutions (including the recent FEDformer and FiLM) focus on long-term forecasting of time series, and their performances on short-term forecasting are quite poor (the Corr metric is less than 0.5 in Table 2 of the main paper). In our humble opinion, for long-term forecasting, we are to learn the general trend and the general periodicity of the time series, and they are relatively easy to learn without tuning hyper-parameters. In fact, a recent paper [1] has shown that a non-parametric baseline model based on periodicity can actually achieve comparable performance to state-of-the-art Transformer-based models on various datasets.
> > >
> > > [1] FreDo: Frequency Domain-based Long-Term Time Series Forecasting. arXiv:2205.12301
> > >
> > > **Q2: Again, the models I mentioned in the first point are usually trained with fewer than 20 epochs (while SCINet using150 epochs). I believe an additional comparison on the training and inference speed will be helpful to support the SCINet's efficiency.**
> > >
> > > A: The original description for this part is kind of misleading, we state in the paer that we train SCINet for at most 150 epochs, but we did not mention the average case. In fact, there is only one case with this number (Exchange dataset under short-term forecasting). For all the long-term forecasting tasks as those in Transformer-based solutions, the number of training epochs is smaller than 20, as shown in the following table.
> > >
> > > |Dataset||||||
> > > |--------------|------------|----|-----|-----|------|
> > > | Exchange     | Horizon    | 96 | 192 | 336 | 720  |
> > > |              | Best Epoch | 16 | 5   | 3   | 1    |
> > > | Elelctricity | Horizon    | 96 | 192 | 336 | 720  |
> > > |              | Best Epoch | 8  | 7   | 8   | 9    |
> > > | Traffic      | Horizon    | 96 | 192 | 336 | 720  |
> > > |              | Best Epoch | 2  | 3   | 2   | 7    |
> > > | ETTh1        | Horizon    | 24 | 48  | 168 |      |
> > > |              | Best Epoch | 6  | 11  | 2   |      |
> > > | ETTh2        | Horizon    | 24 | 48  | 168 |      |
> > > |              | Best Epoch | 2  | 3   | 12  |      |
> > >
> > > Please note that, we have analyzed the computational complexity of SCINet in Section 3.5 of the main paper, which is on par with the conventional TCN architecture and less than Transformer-based solutions. Considering the inputs to time series forecasting models are usually much smaller compared to other applications (e.g., CV and NLP), the inference speed is usually not a concern.
> > >
> > > **Finally, we've noticed that the review score is reduced after the rebuttal (from 6 to 5)**, which seems to be inconsistent to the other part of the review (soundness: 3; presentation: 4; contribution: 3). We certainly hope it is a mistake, but if it's not, could the reviewer point out which part of our paper and/or the rebuttal itself lead to the score reduction? Thanks a lot!

---

### Official Review · Reviewer_gmoU · 2022-07-10

**Rating:** 6
**Confidence:** 4
**Soundness:** 3 good
**Presentation:** 3 good
**Contribution:** 3 good

**Summary:**

The paper proposes to progressively downsampling the sequences into two sub-sequences, and then conduct convolution and interaction operations to study complex temporal dynamics of time series data. Experiments on various time series datasets confirm its significant forecasting accuracy improvements over both existing convolutional and Transformer-based models.

**Questions:**

Please refer the weaknesses. My biggest concern is on the anonymous.

**Strengths And Weaknesses:**

Strengths:
1. The core idea of progressively downsampling the sequences into two sub-sequences to study temporal dynamics is novel.
2. Extensive experiments are performed to confirm the effectiveness.
3. Source code is provided for reproducibility.

Weaknesses:
1. The code (https://anonymous.4open.science/r/SCINet-2588) is not anonymous. The following information is included:
     If you find this repository useful for your work, please consider citing it as follows:
     @article{...author={L**, ...},...
     }.  (checked on 2022/07/10. To keep anonymous, only the first letter of the first author's last name is shown here.)
2. For the experiment results in Table 2 and 4, the N/A and - could be filled by re-implementation or re-training.
3. It seems that SCINet needs more memory usage even compared to Transformer based on Table 3.

---

> ### Author Response · Authors · 2022-08-02
> **Response to Reviewer gmoU**
>
> **Q1: The code (https://anonymous.4open.science/r/SCINet-2588) is not anonymous. The following information is included: If you find this repository useful for your work, please consider citing it as follows: @article{...author={L\**, ...},... }. (checked on 2022/07/10. To keep anonymous, only the first letter of the first author's last name is shown here.)**
>
> A: We thank the reviewer for pointing out this issue. It is a careless mistake: the anonymous link is generated from another public GitHub link and it is anonymous when we submit it.  We have recently updated the codebase of the public one, but we are not aware of the fact that it automatically syncs to the anonymous link in this paper. We have fixed this issue and we apologize for the trouble.
>
> **Q2: For the experiment results in Table 2 and 4, the N/A and - could be filled by re-implementation or re-training.**
>
> A:  Some of the datasets do not provide sufficient information to train with existing methods. For example, the solar-energy dataset does not provide timestamp information, making some Transformer-based methods inapplicable. We shall elaborate more on this issue and add the corresponding results in the revised version, whenever possible.
>
> **Q3: It seems that SCINet needs more memory usage even compared to Transformer based on Table 3.**
>
> A: The memory cost of SCINet is on par with existing Transformer-based solutions based on our experiments. The reason that we do not get results for the Traffic dataset with forecasting horizon 720 is partly due to the fact that our model is trained on a single GPU card with only 12GB of memory. We have further optimized our model to reduce memory usage and the result is updated in the revised paper.

---

### Official Review · Reviewer_9stN · 2022-07-11

**Rating:** 5
**Confidence:** 4
**Soundness:** 3 good
**Presentation:** 3 good
**Contribution:** 2 fair

**Summary:**

In this paper, the authors proposed a novel time-series dependency modeling framework for the long/short-term forecasting task. Specifically, the time series is downsampled into different sub-sequences and resolutions and jointly modeled to produce an enhanced overall representation for the forecasting task.

**Questions:**

1. The paper is based on an assumption: "temporal relations are largely preserved after downsampling into two sub-sequences." However, I don't think this holds true in many real-world scenarios. For example, downsampling may result in two sequences with different trends for a volatile time series, such as stock price. Also, some citations are expected to back this assumption.
2. The overall contribution is a little bit incremental. Stacked blocks and the tree-like design in the SCINet are not something new, though the interactive learning between two sub-sequences is interesting.
3. Page 4, line 155. Can the authors provide more details about how the elements in all the sub-features are ordered?
4. Page 4, line 159. It's noticed that the authors used the fully-connected layers as the prediction layer. However, some constraints after the fully-connected layers are necessary to deal with the real-world situation. For example, for traffic forecasting, only a positive value is expected.
5. Page 5, line 165. Which part of the X is concatenated to $\hat{X}^k$?
6. The main results should include an error bar or the standard deviation. Also, the model should be evaluated by both absolute-value-based and percentage-based metrics.
7. The motivation for adopting intermediate supervision is not well supported. It would be better if the authors could provide more details or demonstrate it through the ablation study.

**Limitations:**

The authors partially pointed out this work's limitations regarding missing data and irregular time series. Other potential limitations are listed in the questions above.

**Strengths And Weaknesses:**

Strengths
+ The idea of interactive learning between downsampled sub-sequences is interesting.
+ Overall, the paper is well organized and written.

Weakness
+ The motivation of this paper is built on the assumption that temporal relations are largely preserved after downsampling into two sub-sequences which may not hold true in many real-world situations such as volatile time series in the financial domain.
+ In the main result, it's better to include the error bars instead of simply providing the prediction results to report the model performance.
+ The design motivation of some critical points, such as intermediate supervision, is not clearly explained or demonstrated in the experiments.
+ The overall innovation is a bit incremental. The major contribution may come from the design of the interactive learning of two sub-sequences. .

---

> ### Author Response · Authors · 2022-08-02
> **Response to Reviewer 9stN**
>
> **Q1: The paper is based on an assumption: "temporal relations are largely preserved after downsampling into two sub-sequences." However, I don't think this holds true in many real-world scenarios. For example, downsampling may result in two sequences with different trends for a volatile time series, such as stock price. Also, some citations are expected to back this assumption.**
>
> A: Forecasting is only possible for those time series that contain certain trends, periodicity, and/or local stability. For such data, downsampling would preserve most of the information based on the Nyquist-Shannon theorem. For those volatile time series without such features, whether they are predictable is in fact debatable.  Considering the stock price, people would buy or sell early as soon as the predicted values are available (if any), making the forecasting at the future timestamps useless.
>
> **Q2: The overall contribution is a little bit incremental. Stacked blocks and the tree-like design in the SCINet are not something new, though the interactive learning between two sub-sequences is interesting.**
>
> A: We agree with the reviewer that the stacked blocks and the tree-like structure are not new even though they are less used in time series modeling, and we have not claimed them to be the contributions of this work.
>
> The main contribution of this work is the SCI-Block design. Compared to the existing temporal convolutional network (TCN) that extracts the average temporal features from the data/features in the previous layer of the network with the shared convolutional filter, our design employs multiple convolutional filters at every layer of the hierarchical network, facilitating the effective extraction of temporal dynamics. This is not possible without sampling sub-sequences first. We have also designed an interactive learning module to allow information interchange between the downsampled sub-sequences. All the above concepts are new for time series modeling and forecasting. With the proposed design, SCINet achieves significant forecasting accuracy improvements over both TCNs and Transformer-based solutions.
>
> **Q3: Page 4, line 155. Can the authors provide more details about how the elements in all the sub-features are ordered?**
>
> A: Since SCINet downsamples the input sequence into the odd and even sub-sequences in the SCI-Block, after processing at each layer, we rearrange them according to their original indices in the input sequence. The details can be seen in our code.
>
> **Q4: Page 4, line 159. It's noticed that the authors used the fully-connected layers as the prediction layer. However, some constraints after the fully-connected layers are necessary to deal with the real-world situation. For example, for traffic forecasting, only a positive value is expected.**
>
> A: As an encoder-decoder architecture for time series modeling, the key part is at the encoder to effectively extract temporal features from the look-back windows. Sophisticated decoder design is not necessary (and often harmful) for temporal relation extraction. For the mentioned scenario, because all the supervised values for the traffic dataset are positive, it is impossible to obtain a negative value in forecasting.
>
> **Q5: Page 5, line 165. Which part of the X is concatenated to $X^k$?**
>
> A: We use the last $T-\tau$ time steps ($T$ is the look-back window length and $\tau$ is the forecasting horizon) in the input X and concatenate them to $X^k$ to make its length equal to $T$.
>
> **Q6: The main results should include an error bar or the standard deviation. Also, the model should be evaluated by both absolute-value-based and percentage-based metrics.**
>
> A: We thank the reviewer for this suggestion and we have added the error bars for the ETTh1 dataset in the revised supplementary material (see Table 1).  Based on our experimental results, the standard deviation is quite small.
>
> We have not added the percentage-based metrics because we are not clear about their exact meaning.
>
> **Q7: The motivation for adopting intermediate supervision is not well supported. It would be better if the authors could provide more details or demonstrate it through the ablation study.**
>
> A: Intermediate supervision is a common strategy to ease the learning procedure for stacked network architectures (e.g., [1,2]). At the same time, as can be seen in Tables 4-6 in the supplementary material, the best results are obtained with only one stack in most cases. In other words, adding more stacks and introducing intermediate supervision provide marginal benefits in a few cases only.
>
> [1] Alejandro Newell, Kaiyu Yang, and Jia Deng, "Stacked Hourglass Networks for Human Pose Estimation", CVPR 2016.
>
> [2] Shaojie Bai, J. Zico Kolter, and Vladlen Koltun, "Trellis Networks for Sequence Modeling", ICLR 2019.

---

> > ### Comment · Reviewer_9stN · 2022-08-09
> > **Thanks for the authors' detailed responses**
> >
> > In general, I'm satisfied with the authors' responses.
> >
> > Regarding Q1. My point is that in a non-autoregressive setup, where multiple time series from different sources are combined as input for a forecasting task (e.g., predicting a company's credit risk using its financial statements and stock price), the assumption may not hold. But I agree with the authors that in this paper's setup, "volatile time series without such features, whether they are predictable is in fact debatable." Thanks for the clarification.
> >
> > Regarding Q4. I doubt that adding a simple constraint layer (e.g., an activation function) would lead to a sophisticated decoder layer. It would be great to see more discussion in the paper.

---

### Meta-Review · Area_Chair_85Vt · 2022-08-27

**Recommendation:** Accept
**Confidence:** Certain

**Metareview:**

Reviewers agree that the proposed idea of interactive learning between two down-sampled sub-sequences is interesting and novel in time-series forecasting. Extensive experiments over 9 datasets and against various baseline methods are conducted to confirm the effectiveness of the proposed SCINet architecture. The paper is well organized and written, and the source code is provided for better reproducibility. Therefore, AC recommends acceptance.

It is worth mentioning that several reviewers raised their concerns towards the overall novelty, as tree-structured framework and stacked modules with intermediate supervision are not new in this area. Authors confirm that the main contribution lies in the proposal of SCI-Block design, which successfully captures long-time dependency even with convolution operations (instead of attention mechanism). This is built upon the assumption that temporal relations are largely preserved after down-sampling into two sub-sequences. However, it remains unclear how to verify whether the time-series forecasting task at hand meets such assumption. AC would like to encourage authors to further discuss this topic in their revised manuscript.

**Award:**

No

---

### Decision · Program_Chairs · 2022-09-14

Accept